# DTR: Towards Optimal Token Compression with Data-driven Token Ranking for Efficient Vision-Language Model Inference

## Abstract

Token compression is crucial for vision-language models (VLMs) inference due to its tremendous computational complexity. Although substantial works with various model-driven methods have been done to mine importance rankings among tokens for compression (e.g., rank according to attention scores or matrix ranks), they are all constrained by one-sided handcrafted information, thus being trapped in local optimum. To utilize comprehensive information for global optimum, we present a Data-driven Token Ranking (DTR) framework, which trains a plug-and-play token-ranking model with self-gathered token-ranking data for VLM token compression at runtime. Specifically, first, we propose a dataset construction method to efficiently gather importance rankings of tokens based on original VLM datasets. Then we present a training method to build a token-ranking model for predicting a ranked-list of token importance based on input vision and text tokens. Finally, the ranking model can be plugged in the model, then filter tokens with an user-defined token number at runtime for acceleration. Extensive experimental results across 8 mainstream benchmarks show that DTR achieves the state-of-the-art token compression performance compared with 8 challenging comparatives. Moreover, a comprehensive analysis shows that DTR as well as data-driven methods possess tremendous potential, which can comprehensively outperform the vanilla VLM with much fewer tokens.

## 1 Introduction

Recent vision-language models (VLMs) have demonstrated impressive capabilities, while their inference efficiency remains a critical bottleneck Jin et al. (2024). A key contributor to this limitation is the excessive number of vision tokens, which often dominate the input sequence to the VLM and lead to substantial computational overhead. This issue is particularly exacerbated in recent high-resolution methods Liu et al. (2024a;b); Chen et al. (2024b); Li et al. (2025) and video-based models Zhang et al. (2024b); Cheng et al. (2024); Varma et al. (2025), where the vision token number grows rapidly with spatial and temporal resolutions. Consequently, such overhead poses a major bottleneck for wildly deploying VLMs in widespread real-world applications.

Existing studies Shang et al. (2024); Chen et al. (2024a); Yang et al. (2025); Zhang et al. (2025b) have explored accelerating VLM inference by reducing vision tokens while preserving essential information. At the core of these approaches lies a common strategy of ***importance-based token ranking***, in which vision tokens are assessed, ranked, and selectively pruned based on their importance—either through implicitly learned attention distributions or explicitly designed heuristics. This process enables efficient inference under computational or user-defined constraints. For instance, VisionZip Yang et al. (2025) performs global selection of dominant vision tokens, while FastV Chen et al. (2024a) prunes tokens dynamically based on attention weights during LLM decoding. SparseVLM Zhang et al. (2025b) introduces a text-guided token pruning framework that directly reuses cross-modal attention maps from decoder layers. TokenCarve Tan et al. (2025) leverages the information quantity measured by the matrix rank of the attention output matrix, and introduces a two-stage token compression framework to achieve aggressive token reduction.

Although these approaches achieve promising results, they typically rely on hand-crafted heuristics or architecture-specific strategies for token ranking. Such **model-driven methods based on human heuristics** are prone to local optima and often struggle to generalize across different domains, which leads to severe degradation on various tasks.

To avoid human heuristics and acquire comprehensive knowledge from various tasks, we learn from the success of current deep learning paradigm and leverage the **data-driven method that extracts high-level knowledge from massive data in an end-to-end manner**. Therefore, instead of any hand-crafted heuristics, we propose a data-driven token-ranking (DTR) framework for token compression, which trains a token-ranking model (TRM) based on self-gathered token-ranking data.

Our main contributions are summarized as follows.

- We propose a data-driven token-ranking framework DTR for token compression. Specifically, a token-ranking dataset construction method and a token-ranking model training method are proposed to train a token-ranking model with self-gathered token-ranking dataset. Then the token-ranking model can be plugged into a VLM for runtime acceleration with user-defined numbers of tokens.

- We conduct extensive experiments and an in-depth analysis to demonstrate the effectiveness of DTR. Generally, DTR achieves the best average performance in all scenarios and works relatively better with fewer tokens. Specifically, compared to 8 comparatives, DTR achieves the best performance in 37/40 scenarios, and up to 325% with average of 53% performance improvement among 8 benchmarks. Moreover, the results of run-time profiling shows that DTR can accelerate the inference in practice. DTR with 8 tokens achieves an average of $7.80\times$ and $2.44\times$ speed-up on the prefilling and the decoding stage across 4 benchmarks, respectively.

- We pioneer a new path for the token ranking problem based on data-driven methods instead of model-driven methods, and conduct a comprehensive analysis to show the tremendous potential of DTR and data-driven path. Specifically, experiments of the DTR's upper-bound performance show that DTR with 32 tokens can even far exceed that of the vanilla VLM with 576 tokens across 8 benchmarks, which reaches an average 29.7% and up to 40.6% performance improvement compared to vanilla VLM. Moreover, experiments of replacing DTR's top-j important tokens with the ground truth demonstrates DTR can steadily improve as the performance of data-driven. Specifically, the performance of replacing DTR's top-2 tokens has surpassed that of vanilla VLM.

## 2 RELATED WORKS

### 2.1 VISION-LANGUAGE MODELS

Large Language Models (LLMs) Vicuna (2023); Touvron et al. (2023); Bai et al. (2023a); Team (2023); Achiam et al. (2023) have achieved great success across a wide range of natural language processing tasks, including text understanding, generation, and question answering. Nonetheless, their reliance on purely textual inputs fundamentally limits their ability to model human-like perception, which is inherently multimodal. This motivation has driven the emergence of vision-language models (VLMs)Liu et al. (2023); Bai et al. (2023b); Chen et al. (2024b); Reid et al. (2024); Li et al. (2024a); Liu et al. (2024e), which integrate visual encoders with LLMs to enable joint understanding of visual and textual information. Typical image and video-based VLMs Liu et al. (2024a); Cheng et al. (2024); Lin et al. (2023) adopt a vision encoder (*e.g.*, ViT Dosovitskiy (2020)) followed by an MLP projection head to align visual features with the LLM's input space, allowing visual instruction tuning for downstream tasks. Despite their effectiveness, this paradigm often requires a large number of visual tokens, especially in high-resolution image or long video scenarios, to preserve fine-grained spatial and temporal information. The resulting increase in token length significantly amplifies computational cost and inference latency, posing a major bottleneck for the scalability and real-time deployment of VLMs in practical applications Jin et al. (2024).

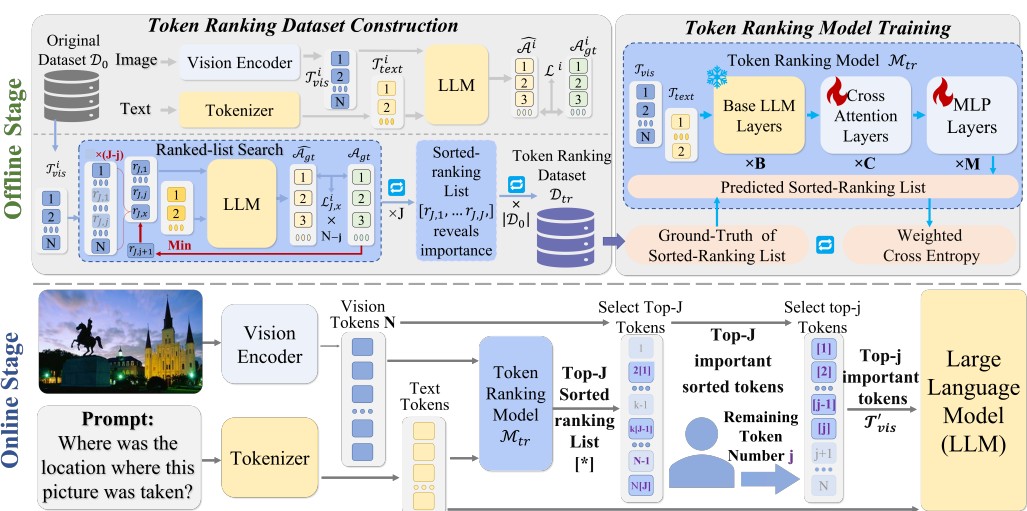

Figure 1: Overview of DTR, which comprises an offline stage for constructing a token ranking dataset from an original dataset and training a token sorting model with a novel task, and an online stage for inference with an user-defined number of tokens.

## 2.2 MODEL-DRIVEN VISUAL TOKEN COMPRESSION

The quadratic complexity of Transformer architectures Vaswani et al. (2017), which scales with the length of input sequences, remains a key bottleneck in vision-language models (VLMs). To alleviate this, several methods Bai et al. (2023b); Cha et al. (2024); Li et al. (2025); Hu et al. (2024); Chu et al. (2024) propose efficient visual projectors that generate compact visual representations using fewer visual tokens before passing them into the LLM. Alternatively, recent works Shang et al. (2024); Chen et al. (2024a); Zhang et al. (2024a; 2025b); Yang et al. (2025); Tan et al. (2025) seek to reduce visual tokens in a training-free manner. LLaVA-PruMerge Shang et al. (2024) exploits sparsity in the visual encoder by analyzing attention distribution between the class token and visual tokens to adaptively prune less informative ones. FasterVLM Zhang et al. (2024a) estimates token importance based on attention scores with the [CLS] token. VisionZip Yang et al. (2025) selects globally dominant tokens through statistical analysis, while FastV Chen et al. (2024a) dynamically prunes tokens during LLM decoding based on attention weights. SparseVLM Zhang et al. (2025b) introduces a text-guided token compression strategy that reuses cross-modal self-attention maps from decoder layers. TokenCarve Tan et al. (2025) proposes a two-stage pruning framework, leveraging information-preservation-guided selection to retain tokens with high information content. While these approaches show encouraging results, they often rely on hand-crafted heuristics or architecture-specific designs, which can lead to suboptimal performance and limited generalization across domains. In contrast, our method adopts a fully data-driven approach, where the token compression policy is learned from large-scale data in an end-to-end manner. This enables more effective, generalizable token compression for VLMs, aiming for globally optimal efficiency–performance trade-offs.

## 3 METHOD

### 3.1 FRAMEWORK OVERVIEW

As illustrated in Fig.1, DTR has offline and online stages.

The offline stage aims to train a Token-Ranking Model (TRM) that takes the multi-modal tokens to predict a ranked-list revealing the importance rankings of visual tokens. Specifically, based on an user-defined maximum number of remaining tokens $J$, the token-ranking dataset construction process expands an original VLM dataset to a token-ranking dataset with token-ranking labels from $[1, 2, ..., J + 1]$, where 1 to $J$ reveals the relative importance among remaining tokens and $J + 1$ represents the unimportant tokens (see Sec.3.2). Then, based on the token-ranking dataset, token-

ranking model training process trains a TRM to predict a ranked-list $\boldsymbol{r}_J^i = [r_{J,1}^i, ..., r_{J,J}^i]$ for token ranking and compression at runtime (see Sec.3.3).

The online stage aims to run a VLM with a TRM for efficient inference. Specifically, it first inserts the TRM into a selected compression layer of VLM as a plug-and-play module, then filters out intermediate vision tokens based on an user-defined number of remaining tokens $j$ and a ranked-list predicted by TRM $\boldsymbol{r}_J^i$ at runtime to accelerate the remaining VLM inference (see Sec.3.4).

## 3.2 TOKEN-RANKING DATASET CONSTRUCTION

Token-ranking dataset construction aims to expand a given VLM dataset $\mathcal{D}_0$ to a token-ranking dataset $\mathcal{D}_{tr}$ for the supervised end-to-end training of token-ranking model (TRM), which can be formulated as follows.

$$\mathcal{D}_{tr} = \{\{\mathcal{T}_{vis}^i, \mathcal{T}_{text}^i, \mathcal{R}^i\}, i = 1, 2, ..., |\mathcal{D}_0|\}. \tag{1}$$

Here, $\mathcal{T}_{vis}^i$ and $\mathcal{T}_{text}^i$ are paired image and text tokens, which can be considered as inputs of a certain intermediate layer in VLM (e.g., inputs to the LLM of VLM). Assume $\mathcal{T}_{vis}^i$ consists $N$ vision tokens, the ranked-list set $\mathcal{R}^i$ is as follows.

$$\mathcal{R}^i = \{\boldsymbol{r}_j^i = [r_{j,1}^i, r_{j,2}^i.., r_{j,j}^i], j = 1, 2.., J\}, \tag{2}$$

where $1 \leq J \leq N$ is an user-defined maximum number of remaining tokens. $\boldsymbol{r}_j^i$ represents the ranked-list for the top-j important tokens in $\mathcal{T}_{vis}^i$, where $r_{j,k}^i$ is the index of $k_{th}$ important token in the ranked-list of top-j important tokens.

To build such a dataset, there exist two key questions.

**First, how to determine the importance of a token set**? Instead of the loss of hand-crafted features (e.g., the attention scores Zhang et al. (2025b) or the matrix rank Tan et al. (2025) of intermediate layers), the end-to-end loss $l_j^i$ of VLM reveals the final task performance with compressed input tokens $\{\mathcal{T}_{\boldsymbol{r}_j^i}, \mathcal{T}_{text}^i\}$, thus it can be considered as a most effective proxy of importance based on the posterior knowledge. The smaller the loss, the more important the token set is.

**Second, how to get the optimal ranked-list set?** Intuitively, the traversal method achieves the optimum with guarantee. However, the computing complexity of this method is $O(N!)$, which is unaffordable in practice. For example, considering the minimum number of remaining tokens after compression in existing works (e.g., remaining 64 of total 576 for LLaVA-1.5 series in existing works), the traversal method requires $576! - 512! \approx 1.22 \times 10^{175}$ times forward VLM inferences for an optimal ranked-list $\boldsymbol{r}_{64}^i$. Though this process is laborious, unfortunately, when the number of remaining tokens changes, this process needs repeating from scratch (e.g., $\boldsymbol{r}_{63}^i$ still needs $2.39 \times 10^{172}$ times searching with a known $\boldsymbol{r}_{64}^i$).

To reduce computing complexity and improve reusability for $\boldsymbol{r}_j^i$, we present a ranked-list search algorithm with one-step greedy strategy, whose pseudo is given in Alg.1.

Based on the one-step greedy strategy which reuses the existing optimal ranked-list and only search the next important token, the computing complexity of this algorithm is only $O(N^2)$, which can decrease the searching number from $1.22 \times 10^{175}$ to $3.48 \times 10^4$ for $\boldsymbol{r}_{64}^i$. Furthermore, due to the single-step recursion in Alg.1 (i.e., $\boldsymbol{r}_{j-1}^i = \boldsymbol{r}_j^i[: -1]$), the whole ranked-list set $\mathcal{R}^i$ can be represented by $\boldsymbol{r}_J^i$ (i.e., $\mathcal{R}^i \cong \boldsymbol{r}_J^i$), and the total searching times of a whole ranked-list set $\mathcal{R}^i$ is equal to that of $\boldsymbol{r}_J^i$ (i.e., $\frac{J(2N-J+1)}{2}$ times).

## 3.3 TOKEN-RANKING MODEL TRAINING

Token-ranking Model (TRM) Training aims to train a model that takes multi-modal tokens from an intermediate layer of a VLM and outputs a Top-J ranked-list revealing the Top-J importance rankings of input multi-modal tokens. There exist two key questions of this module.

**First, what is the architecture of this model**? There exist three choices with different concerns.

Intuitively, a small stacked transformer model can be used to directly fit the training data from scratch and predict the ranked-list for test samples. This simple model can achieve the most efficiency with

---

**Algorithm 1:** Ranked-list Search for $i_{th}$ Sample

---

**Input:**
$\mathcal{T}^i_{vis} = (\boldsymbol{t}_1, ..., \boldsymbol{t}_N)$: vision tokens of $i_{th}$ sample ;
$\mathcal{T}^i_{text}$: text tokens of $i_{th}$ sample;
$\mathcal{M}(\mathcal{T})$: VLM inference with tokens $\mathcal{T}$;
$\mathcal{L}^i_{gt}(\mathcal{M}(\mathcal{T}), \mathcal{A}^i_{gt})$: loss function with ground-truth;
**Output:**
$\mathcal{R}^i = (\boldsymbol{r}^i_1, ..., \boldsymbol{r}^i_J)$: optimal ranked-list set.

---

**1** $\boldsymbol{r}^i_0 = [], \{\mathcal{L}_j = [], j = 1, 2, ..., J\}$ %    *Initialize*         %
**2 for** $j = 1$ **to** $J$ **do**
**3**     **for** $\boldsymbol{t}_x$ *in* $\mathcal{T}^i_{vis}$ **do**
**4**        $\mathcal{L}_j$.append($\mathrm{L}^i_{gt}(\mathcal{M}(\mathcal{T}_{\boldsymbol{r}^i_{j-1}} + \boldsymbol{t}_x, \mathcal{T}^i_{text}), \mathcal{A}^i_{gt})$)
       *% Store the losses with Top -(j-1) tokens and all optional $j_{th}$ important tokens*    %
**5**     **end**
**6**     $\boldsymbol{R}^i_{j*} = [R^i_{j,1}, .., R^i_{j,N}] \leftarrow \mathrm{Sort}(\mathcal{L}_j)$
    *% Get a single ranked-list of the $j_{th}$ important token according to sorted losses*    %
**7**     $\boldsymbol{r}^i_j = [\boldsymbol{r}^i_{j-1}, r^i_{j,j} = R^i_{j,1}], \mathcal{T}^i_{vis} \leftarrow \mathcal{T}^i_{vis} - \boldsymbol{t}_{r^i_{j,1}}$;
    *% Get the ranked-list of top-j important tokens $\boldsymbol{r}^i_j$ based on one-step greedy strategy and deleted the*
      *choosen token in $\mathcal{T}^i_{vis}$*    %
**8 end**
**9 return** $\mathcal{R}^i = (\boldsymbol{r}^i_1, ..., \boldsymbol{r}^i_J)$

---

a few parameters. Lacking prior knowledge, this model cannot fully interpret multimodal tokens, limiting its task generalization.

Take a step further, an encoder-decoder model with pretrained weights can be used to handle this fitting task, specifically mapping input tokens to output ranking sequences. However, suffering from the semantic misalignment with encoder-only architectures, this model is not suitable for dominant VLMs with encoder-only architectures, thus it is not selected in this work.

Finally, considering the model adaptability and the training convenience, the encoder-only model with pertained weights is selected. Specifically, LLMs with same series but smaller sizes compared to the LLM of VLM can be selected as the base token-ranking model for model efficiency, semantic alignment and task generalization. Here, we select the first B×LLM layers to utilize the prior knowledge from massive pretraining data, then stacked C×cross-attention layers between $N$ vision tokens and all text tokens to generate $N$ output tokens, finally M×MLP layers for each output token to get a probability distribution of its rankings. The whole model architecture is illustrated in Fig.1, which takes $\mathcal{T}^i_{vis}$ and $\mathcal{T}^i_{text}$ as input, and output the ranked-list $\boldsymbol{r}^i_J$. It is worth noting that the predicted $p_{th}$ important token $r^i_{J,p} = R^i_{p,1}$ maybe the same as the previous $q_{th}$ $(q < p)$ important token $r^i_{J,q}$. When this happens, we make $r^i_{J,p} = R^i_{p,p^*}$, where $p^*$ is the smallest value that $R^i_{p,p^*}$ is not equal to any $q_{th}(q < p)$ important token $r^i_{J,q}$.

**Second, how to train this model for token ranking task**? The model ranks multimodal tokens from a VLM by importance. Based on the training data constructed in Sec.3.2, we can train this model with some certain supervisions. The only question is how to design a loss function that accurately and comprehensively reflects the differences among various tokens. Instead of directly regress a final ranked-list $\boldsymbol{r}^i_J$, we take a more comprehensive task for training.

Specifically, considering the output of $\mathcal{M}_r$ contains N discrete distributions with support size $(J+1)$, the ground-truth ranked-list $\boldsymbol{r}^i_J = [r^i_{J,1}, .., r^i_{J,J}]$ is transferred to $N$ simplified labels with $J + 1$ dimensions, $\mathcal{Y} = [\boldsymbol{y}^i_1, ..., \boldsymbol{y}^i_n, ..., \boldsymbol{y}^i_N]$, where $\boldsymbol{y}^i_n$ can be calculated as follows.

$$\boldsymbol{y}^i_n = [I^i_{n,1}, ..., I^i_{n,j}, ..., I^i_{n,J}, I^i_{n,J+1}],$$
$$\text{where } I^i_{n,j} = \begin{cases} 1 & \text{if } n = r^i_{Jj} \\ 0 & \text{else.} \end{cases} \tag{3}$$

Here, $\boldsymbol{y}_n^i$ indicates that the $n_{th}$ tokens is the $j_{th}$ important token if $I_{n,j}^i = 1$. If $I_{n,1}^i = I_{n,2}^i =, ..., = I_{n,J}^i = 0$, then $I_{n,J+1}^i = 1$ means $n$ does not belong to the ranked-list of top-J important tokens $\boldsymbol{R}_J^i$, therefore we create a new category J+1 for it. Sometimes, a (J+1)-dimensional label may be hard for learning when J becomes large, there exists a simpler label $\boldsymbol{y}_i^n = [\sum_{j=1}^J I_{n,j}^i, I_{n,J+1}^i]$ for replacement, which relaxes the task from J+1 categories to 2 categories.

Then for the prediction of TRM, $\mathcal{M}_r(\mathcal{T}_{vis}^i, \mathcal{T}_{text}^i) = \hat{\mathcal{Y}}^i = [\hat{\boldsymbol{y}}_1^i, ..., \hat{\boldsymbol{y}}_N^i]$, a weighted cross-entropy loss function CE() is adopted for model training as follows.

$$\underset{\mathcal{M}_r}{\arg\min} \frac{1}{N|\mathcal{D}_{tr}|} \sum_{i=1}^{|\mathcal{D}_{tr}|} \sum_{n=1}^{N} (w_n^i \mathrm{CE}(\boldsymbol{y}_n^i, \mathrm{Softmax}(\hat{\boldsymbol{y}}_n^i))),$$

$$\text{where } w_n^i = \begin{cases} \frac{N-J}{J} & \text{if } \hat{\boldsymbol{y}}_n^i[-1] \neq 1) \\ 1 & \text{else.} \end{cases} \tag{4}$$

Here, weight $w_n^i$ is used to mitigate the imbalance between the number of zero and non-zero elements in the distribution.

### 3.4 Model Insertion and Token Filtering

In the online stage, the token-ranking model can be inserted into a compression layer of VLM and works at runtime.

For each input sample, first, the inference keeps the same before the compression layer, which outputs the vision and text tokens $\{\mathcal{T}_{vis}, \mathcal{T}_{text}\}$ for the token-ranking model. Then the token-ranking model with comprehensive prior knowledge assesses vision tokens with text tokens, and outputs a ranked-list $\boldsymbol{r}_J^i$ representing the Top-J important tokens, where a ranked-list for any Top-j import token can be get from $\boldsymbol{r}_J^i$ as $\boldsymbol{r}_J^i[:j] = (r_1^i, ..., r_j^i), 1 \leq j \leq J$.

After that, based on a runtime user-defined number $j$ of remaining tokens, the corresponding tokens $\mathcal{T}_{\boldsymbol{r}_J^i[:j]}$ are selected at runtime as the input to the next layer, then the token number decreases from $N$ to $j$ for the following inference.

## 4 Experiment

### 4.1 Benchmarks and Comparatives

To validate the effectiveness, we conduct comprehensive experiments between DTR and 8 existing methods (i.e., FastV Zhang et al. (2024a), SparseVLM Zhang et al. (2025b), MustDrop Liu et al. (2024c), VisPruner Zhang et al. (2025a), TokenCarve Tan et al. (2025), Random Selection (Random), Uniform Selection (Uniform) and VisionZip Yang et al. (2025)) on 8 widely used benchmarks including MME Fu et al. (2024), MMBench (MMB) Liu et al. (2024d), POPE Li et al. (2023), SQA$^{\text{IMG}}$ Lu et al. (2022), VizWiz Bigham et al. (2010), VQA$^{\text{V2}}$ Goyal et al. (2017), GQA Hudson & Manning (2019) and SEED-Bench (SEED$^{\text{IMG}}$) Li et al. (2024b). It is worth noting that the minimum number of remaining tokens is only 64 in almost all existing works, so we reproduce some challenging methods (i.e., VisPruner, TokenCarve, VisionZip) for fewer tokens. Also worth noting, Random and Uniform methods are added as the baselines. Methods with a poorer performance than the baselines can be considered as incapable of the task. More details can be found in Appendix.

### 4.2 Implementation Details

Considering LLaVA-1.5-7B is the most widely used in mainstream works, we implement DTR on LLaVA-1.5-7B to make a fair and comprehensive comparison.

Specifically, to approximate the compression settings in mainstream works as closely as possible, the compression layer is set as the input layer of the LLM (i.e., Vicuna-7B) in VLM (i.e., LLaVA-1.5-7B), and the compression token set is chosen as all vision tokens in the compression layer. Then, for token-ranking dataset construction, 203,507 single-round conversation samples are selected from the instruction tuning dataset of LLaVA-1.5 for training. For each sample, we set the maximum number

Table 1: Evaluation across eight benchmarks under various vision token compression levels. The bold text denotes the best performance, and the underline denotes the second best. Generally, DTR achieves the best performance in all scenarios except for retaining 32 tokens of Vispruner on VizWiz, 16 and 4 tokens of TokenCarve on SQA$^{\text{IMG}}$. DTR even with 32 remaining tokens achieves better performance than all comparatives with 64 remaining tokens on POPE and SQA$^{\text{IMG}}$ benchmarks. Specifically, compared to VisPruner, TokenCarve, VisionZip, Random and Uniform, DTR achieves up to 92%, 325%, 205%, 81%, 187% performance improvement, respectively. On MME, MMB, POPE, SQA$^{\text{IMG}}$, VizWiz, VQA$^{\text{V2}}$, GQA and SEED$^{\text{IMG}}$, DTR achieves up to 51%, 121%, 325%, 7%, 8%, 52%, 26%, 44% performance improvement than comparatives, respectively. It is worth noting that the relative performance of DTR compared to all comparatives becomes better with remaining tokens decreasing. DTR achieves the consistently best performance across all comparatives and benchmarks with 1 token.

| Method | MME | MMB | POPE | SQA$^{\text{IMG}}$ | VizWiz | VQA$^{\text{V2}}$ | GQA | SEED$^{\text{IMG}}$ | Avg. |
|---|---|---|---|---|---|---|---|---|---|
| **Retain 576 Tokens** (100.0%) | | | | | | | | | |
| **Vanilla-576** | 1868 | 64.6 | 86.1 | 69.5 | 50.0 | 78.5 | 62.0 | 66.2 | 100.0% |
| **DTR$_{\text{ub}}$-32** | **2627** | **83.2** | **97.9** | **87.6** | - | - | **80.1** | **92.7** | **129.7%** |
| **Retain 64 Tokens** (↓ 88.9%) | | | | | | | | | |
| **FastV** | 1564 | 61.0 | 59.2 | **69.9** | 51.8 | 66.3 | 46.1 | 43.7 | 84.5% |
| **SparseVLM** | 1505 | 56.2 | 75.1 | 62.2 | 50.1 | 68.2 | 53.8 | 52.2 | 87.1% |
| **MustDrop** | 1641 | 60.0 | – | 63.4 | 51.2 | 69.3 | 53.1 | – | 91.4% |
| **VisPruner** | 1681 | 61.3 | **80.4** | 69.1 | **53.3** | 72.7 | 55.4 | 58.2 | 94.3% |
| **TokenCarve** | **1754** | **62.0** | 79.9 | 69.7 | 51.4 | **74.8** | 57.7 | 61.4 | **95.8%** |
| **VisionZip** | 1690 | 60.1 | 77.0 | 69.0 | – | 72.4 | 55.1 | 57.8 | 90.1% |
| **Retain 32 Tokens** (↓ 94.4%) | | | | | | | | | |
| **VisPruner** | 1581 | 58.4 | 72.7 | 69.2 | **53.0** | 67.7 | 52.2 | 53.6 | 89.6% |
| **TokenCarve** | 1595 | 59.5 | 69.5 | 69.1 | 50.1 | 69.8 | 53.7 | 56.4 | 89.8% |
| **VisionZip** | 1557 | 57.9 | 68.5 | 69.0 | – | 67.0 | 51.8 | 53.0 | 85.8% |
| **Random** | 1386 | 47.3 | 68.6 | 65.6 | 51.0 | 63.5 | 52.1 | 53.5 | 83.7% |
| **Uniform** | 1508 | 52.4 | 69.0 | 66.3 | 52.0 | 64.4 | 52.3 | 53.8 | 86.1% |
| **DTR(ours)** | **1679** | 59.6 | **82.5** | **70.2** | 51.4 | **71.4** | 54.6 | **60.2** | **94.0%** |
| **Retain 16 Tokens** (↓ 97.2%) | | | | | | | | | |
| **VisPruner** | 1347 | 47.9 | 57.0 | 66.9 | 48.1 | 59.6 | 47.0 | 47.6 | 78.6% |
| **TokenCarve** | 1305 | 47.9 | 49.7 | **70.0** | 47.2 | 59.2 | 47.3 | 48.0 | 77.6% |
| **VisionZip** | 1371 | 50.7 | 53.0 | 67.3 | – | 59.2 | 46.5 | 46.9 | 75.9% |
| **Random** | 1311 | 40.5 | 52.4 | 65.7 | 49.8 | 57.4 | 49.1 | 51.6 | 77.3% |
| **Uniform** | 1303 | 46.7 | 56.8 | 65.5 | 50.3 | 58.3 | 48.6 | 49.4 | 78.8% |
| **DTR(ours)** | **1560** | **53.6** | **77.6** | 68.3 | **50.5** | **67.5** | **51.8** | **57.9** | **89.1%** |
| **Retain 8 Tokens** (↓ 98.6%) | | | | | | | | | |
| **VisPruner** | 1168 | 33.9 | 38.4 | 65.3 | 47.1 | 50.3 | 42.0 | 41.9 | 67.9% |
| **TokenCarve** | 1035 | 31.2 | 26.8 | 66.2 | 45.1 | 48.2 | 41.4 | 39.5 | 63.5% |
| **VisionZip** | 1084 | 28.7 | 24.2 | 64.8 | – | 46.9 | 39.9 | 39.4 | 58.2% |
| **Random** | 1178 | 34.5 | 48.8 | 64.5 | 46.1 | 54.3 | 45.0 | 45.5 | 71.1% |
| **Uniform** | 1159 | 34.1 | 41.2 | 64.4 | 47.9 | 51.3 | 43.7 | 43.7 | 69.1% |
| **DTR(ours)** | **1431** | **45.5** | **73.8** | **67.6** | **48.6** | **63.1** | **48.7** | **54.5** | **83.6%** |
| **Retain 4 Tokens** (↓ 99.3%) | | | | | | | | | |
| **VisPruner** | 1102 | 27.3 | 35.9 | 63.7 | 45.7 | 45.7 | 40.2 | 40.1 | 63.7% |
| **TokenCarve** | 903 | 21.2 | 16.3 | **65.5** | 45.0 | 42.0 | 38.8 | 35.8 | 56.8% |
| **VisionZip** | 926 | 18.1 | 25.2 | 63.6 | – | 38.9 | 36.6 | 35.3 | 51.5% |
| **Random** | 1041 | 24.9 | 38.3 | 64.6 | 46.1 | 46.6 | 41.9 | 44.4 | 64.7% |
| **Uniform** | 1019 | 23.5 | 24.1 | 63.6 | 44.7 | 43.7 | 38.8 | 38.6 | 59.6% |
| **DTR(ours)** | **1360** | **39.9** | **69.2** | 65.3 | **47.5** | **59.2** | **46.1** | **50.9** | **78.8%** |
| **Retain 1 Token** (↓ 99.8%) | | | | | | | | | |
| **VisPruner** | 975 | 21.1 | 44.5 | 63.8 | 44.5 | 41.1 | 37.7 | 37.0 | 60.8% |
| **TokenCarve** | – | – | – | – | – | – | – | – | – |
| **VisionZip** | 881 | 19.1 | 26.7 | 63.3 | – | 38.3 | 36.0 | 34.5 | 51.1% |
| **Random** | 978 | 21.7 | 43.7 | 63.9 | 44.7 | 42.8 | 39.0 | 37.8 | 61.6% |
| **Uniform** | 967 | 21.6 | 43.2 | 63.6 | 44.6 | 42.2 | 38.0 | 37.9 | 61.1% |
| **DTR(ours)** | **1167** | **25.1** | **58.9** | **64.5** | **46.2** | **51.0** | **41.2** | **44.3** | **69.2%** |

*The project of TokenCarve can not be implemented for retaining 1 token, so there exist no results.

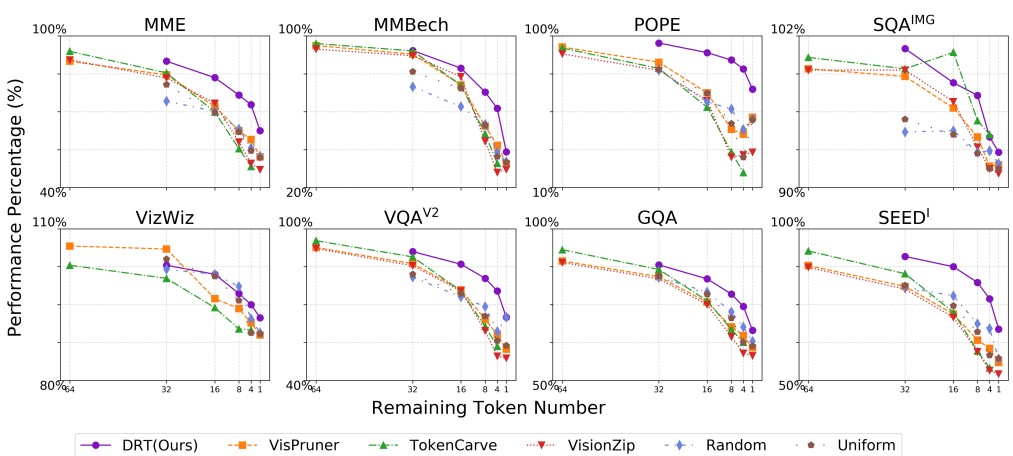

Figure 2: The trends of performance when remaining tokens decrease from 64 to 1.

of remaining tokens $J = 32$ and get a ranked-list set $r_{32}^i$ through Alg.1. Next, for token-ranking model training, the base ranking model can be set as the LLM of LLaVA-7B (i.e., Vicuna-7B), then we set B=10 for the base ranking model, C=2 for cross-attention layers and M=2 for MLP layers.

More details and codes can be found in Appendix.

### 4.3 MAIN RESULTS

Tab.1 illustrates the results of DTR and 8 comparatives among 8 benchmarks. Obviously, compared with existing methods, DTR achieves the best average performance in all scenarios. Generally, compared with existing methods, DTR achieves the best performance in 37/40 scenarios except for 3 scenarios (i.e., 32 tokens of Vispruner on VizWiz, 16 and 4 tokens of TokenCarve on SQA$^{\text{IMG}}$).

To make an intuitive demonstration, taking the Vanilla-576 (i.e., the original LLaVA-1.5-7B with 576 tokens) as the baseline, the trends of all methods are clearly illustrated in the Fig.2. Obviously, DTR outperforms comparatives in most scenarios with reasonable curves of performance percentage-token number (i.e., monotonic curves). However, the curves of other comparatives do not follow this monotonicity, which means a strange phenomenon that the less token achieves the better performance. Here, we analyze that this anomaly arises from interference among tokens: during ranking, predicting an inappropriate token can actually degrade performance. Moreover, surprisingly, some comparatives are even weaker than Random or Uniform in some scenario (e.g., all comparatives on SEED$^{\text{IMG}}$ when token number decreases to 16), which means that these comparatives does not work on these scenarios. In contrast, DTR performs well across all scenarios, which shows its good generalization from its data-driven learning paradigm.

### 4.4 ANALYSIS AND DISCUSSION

**The upper-bound performance of DTR**. Though DTR achieves state-of-the-art performance across tasks, its upper performance bound remains unclear. Such analysis can guide further improvements and highlight the promise of data-driven token compression.

Therefore, we utilize the ranked-list search in Alg.1 on various benchmarks to get their ground-truths of ranked-lists, then make inference with these ranked-lists instead of DTR's prediction. This performance can be considered as the upper bound of DTR, which implies that DTR can fully fits ground-truths. The results among different benchmarks are illustrated in Tab.1, where the upper-bound performance of DTR with 32 remaining tokens DTR$_{\text{ub}}$ is extraordinarily outstanding. Though with much fewer tokens, DTR$_{\text{ub}}$ vastly surpasses the vanilla model with 576 tokens Vanilla-576 in all benchmarks. Specifically, DTR$_{\text{ub}}$ achieves an average 29.7% and up to 40.6% performance im-

provement compared to Vanilla-576. These promising results indicate that DTR still has enormous room for improvement and can achieve less but better performance, which means not only acceleration but also precision enhancement.

For a comprehensive analysis of DTR's potential gains, we substitute the top-j tokens in its Top-32 ranked list with ground-truth labels, thereby simulating a more accurate DTR. (j=1,2,4,8,16, $DTR_{re}$-j in Tab.5). Tab.5 illustrates the performance gains with additional replacements, which indicates the potential improvement of DTR. An exciting finding is that $DTR_{re}$-2 has surpassed the original model Vanilla-576, which shows the considerable potential of DTR in the near future.

Table 2: The improvement of DTR by replacing predicted ranked-lists with some GT labels.

| Tasks | $DTR_{re}$-0 | $DTR_{re}$-1 | $DTR_{re}$-2 | $DTR_{re}$-4 | $DTR_{re}$-8 |
|---|---|---|---|---|---|
| **POPE** | 82.5 | 85.8 | 87.0 | 88.2 | 89.5 |
| **SQA$^{IMG}$** | 70.2 | 70.9 | 72.0 | 73.5 | 75.3 |

**Profiling of runtime performance**. As the purpose of token compression is runtime acceleration, latencies on prefilling and decoding stage are profiled to show the runtime performance of DTR. Here, all experiments are conducted on a NVIDIA A40 GPU and the batch size is set as the highest affordable value of vanilla model (Vanilla-576 in Tab.3) for a fair comparison, though DTR can make inference with higher batch size and achieve better runtime performance. Tab.3 illustrates the impressive runtime performance of DTR with top-j remaining tokens (DTR-32/16/8), which achieves up to $7.80\times$ and $2.44\times$ speed-up on the prefilling and the decoding stage, respectively. Moreover, with the same batch size, DTR only consumes $11.25\%\sim15.00\%$ GPU memory for KV-Cache compared with that of Vanilla-576 in inference, which shows the memory optimization.

Table 3: Runtime latencies of the prefilling/decoding stage with different methods and benchmarks.

| Task | Vanilla-576 | DTR-32 | DTR-16 | DTR-8 |
|---|---|---|---|---|
| **MME** | 100.4/3.6 | 16.2/1.4 | 14.2/1.4 | 12.6/1.4 |
| **POPE** | 98.5/3.4 | 15.3/1.4 | 13.1/1.4 | 11.1/1.3 |
| **VizWiz** | 114.0/3.2 | 21.3/1.5 | 19.1/1.5 | 17.3/1.5 |
| **VQA$^{V2}$** | 98.7/3.1 | 15.7/1.4 | 13.2/1.3 | 11.7/1.3 |
| **Avg.** | 102.9/3.3 | 17.1/1.4 | 14.9/1.4 | 13.2/1.4 |

**Overhead of DTR**. Though DTR achieves state-of-the-art acceleration with low memory consumptions, due to the extra token-ranking model, there exists a certain overhead of DTR. In practice, the DTR model consumes approximately 4,692.41 MB of GPU memory and incurs an average extra latency of 15 ms for a single inference. It is worth noting, there exist many controllable settings of DTR architecture, which can efficiently reduce the overhead. With these settings, users can control the trade-off between the accuracy and efficiency according to their demands.

More analyses can be found in Appendix.

## 5 CONCLUSIONS

Token compression relies on the token ranking to access the importance of tokens and prune the relative unimportant tokens. However, existing works based on model-driven methods (e.g., attention scores or matrix ranks) perform poorly in learning such a complex nonlinear relationship. Therefore, in this work, we present a data-driven token ranking framework (DTR) to learn the complex mapping between input tokens and its importance rankings from massive self-gather token-ranking data in an end-to-end manner. Specifically, first, we propose a dataset construction method to gather the importance rankings of tokens. Then we present a training method to train a token-ranking model for predicting a ranked-list of token importance based on input tokens. Finally, as a plug-and-play module in VLM, the token-ranking model filters tokens with a user-defined number at runtime for acceleration. Experimental results across 8 mainstream benchmarks show that DTR achieves the state-of-the-art compared with 8 challenging comparatives and data-driven methods hold tremendous potential for token compression.

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

## APPENDIX

In this document, additional experimental results and more details on DTR are provided. The supplementary material is organized as follows:

- § A: Details of Benchmarks;
- § B: Details of Implementations;
- § C: Analysis of Token-ranking Dataset;
- § D: Analysis of Token-ranking Model;
- § E: Qualitative Analysis of DTR;

Furthermore, we will fully release the **source code and checkpoints**. We provide example code files along with this document, including the inference of LLaVA-1.5-7B with DTR across various benchmarks and the reproduction of challenging comparatives (i.e., VisPruner Zhang et al. (2025a), Token Carve Tan et al. (2025), VisionZip Yang et al. (2025)) in supplementary materials. An intuitive comparison is illustrated in the Fig.3.

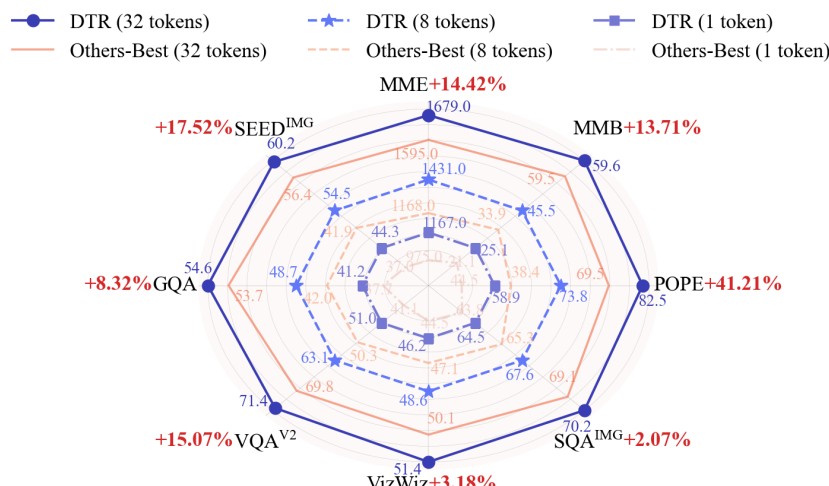

Figure 3: Performance of DTR on mainstream benchmarks when compressing visual tokens of LLaVA-1.5-7B. Others-Best comes from the best performance of three challenging comparatives (i.e.,Vispruner, TokenCarve, VisionZip)

## A  DETAILS OF BENCHMARKS

The details of benchmarks used in the main text are summarized in Fig.4 to show their tasks and measurements. POPE Li et al. (2023) and MME Fu et al. (2024) are dedicated hallucination-oriented benchmarks, as reported in the main text, DTR achieves state-of-the-art performance on both benchmarks, indicating its strong discriminative power against object-level and attribute-level hallucinations. SEED[IMG] Li et al. (2024b) consists of 12-dimension multi-task data. DTR also achieves the best performance in this benchmark, evidencing the robust multi-task capability of our model. Similarly, DTR still performs well on MMB Liu et al. (2024d) and VQA[V2] Goyal et al. (2017), demonstrating its robust performance in both multi-choice and open-ended scenarios. GQA Hudson & Manning (2019) and SQA[IMG] Lu et al. (2022) are reasoning-heavy benchmarks where our model performs well but not the best in 16 and 4 tokens, this indicates that there still needs more efforts for complex reasoning tasks. VizWiz Bigham et al. (2010) can be seen as a simple image benchmark. Not only DTR, but also challenging comparatives (i.e., VisPrunerZhang et al. (2025a), TokenCarveTan et al. (2025), VisionZip Yang et al. (2025)) perform well in this benchmark, which seems that this benchmark is incapable of adequately reflecting the differentiation. We think more challenging benchmarks should be designed in the future.

| Benchmark Task | Description | Measurement | Example | |
|---|---|---|---|---|
| MME

Perception, Cognition | Multimodal Model Evaluation tests the performance of perception and reasoning via concise yes/no questions about image–text co-occurrence facts | The sum of perception and cognition accuracy obtained from LLaVA evaluation scripts | ```
a = 10
b = 100
c = b * a
print(a, b, c)
``` | **Question**: Is a python code shown in the picture?\nAnswer the question using a single word or phrase.
**Answer**: Yes |
| MMB

Fine-grained multi-choice | MM-Bench gauges overall multi-modal competence across twentyplus fine-grained task dimensions using a large-scale visual–textual item bank | The overall score obtained from online Open Compass evaluation | | **Question**: Is plastic a mineral. Plastic has the following properties: solid, no fixed crystal structure, not a pure substance, made in a factory. A yes, B no.
**Answer**: B |
| POPE

Hallucination detection | Polling-based Object Probing Evaluation checks robust object-presence judgments in complex scenes through binary yes/no polling | The average F1 scores obtained from LLaVA evaluation scripts | | **Question**: Is there a snowboard in the image?\nAnswer the question using a single word or phrase.
**Answer**: yes |
| SQAIMG

Science diagram QA | ScienceQA-Image subset draws image-coupled questions from grade-school science exams to test visual-evidence–based scientific reasoning. | The image accuracy from LLaVA evaluation scripts | | **Question**: What is the name of the colony shown?\nA. Maryland\nB. New Hampshire\nC. Rhode Island\nD. Vermont
**Answer**: B |
| VizWiz

Real-world low-quality images | VizWiz Grand Challenge benchmarks real-world usefulness by asking questions on blurry, poorly lit photos taken by visually impaired users seeking help | The overall score obtained from online EvalAI | | **Question**: What is this? and what color is it?\nWhen the provided information is insufficient, respond with 'Unanswerable'.\nAnswer the question using a single word or phrase.
**Answer**: \ |
| VQAV2

Open-ended COCO QA | Visual Question Answering v2 presents over a million open-ended questions on natural images to measure fine-grained visual understanding and answer accuracy | The overall score obtained from online EvalAI | | **Question**: What credit card company is on the banner in the background?\nAnswer the question using a single word or phrase.
**Answer**: \ |
| GQA

Compositional relational QA | Generalized Question Answering leverages scene-graph-derived multi-hop queries to assess compositional reasoning and explainability over structured visual knowledge | The accuracy obtained from LLaVA evaluation scripts | | **Question**: Who is wearing the dress?\nAnswer the question using a single word or phrase.
**Answer**: woman |
| SEEDIMG

12-dimension multi-task | SEED-Bench Image subset supplies standardized, scalable image-level evaluation across twelve carefully curated capability dimensions for large multimodal models | The image accuracy from LLaVA evaluation scripts | | **Question**: How many towels are in the image?\nA. One\nB. Two\nC. Three\nD. Four\nAnswer with the option's letter from the given choices directly.
**Answer**: A |

Figure 4: Details of benchmarks.

## B   DETAILS OF IMPLEMENTATIONS

**Architecture of token-ranking model.** Fig.5 illustrates the detailed architecture of token-ranking model (TRM). The token-ranking model first processes both image and text tokens through B layers of an base ranking model (i.e., pretrained LLM layers from the same series but smaller LLM of VLM). These intermediate tokens are then passed through C cross-attention layers, where both image and text tokens serve as keys and values, enabling the model to progressively fuse information across modalities and generate merged image tokens. Finally, the merged image tokens are fed into M MLP layers to reduce their dimensionality, yielding the final prediction of the distribution.

**Training settings of token-ranking model.** The token-ranking model is trained for 200 epochs using a cosine-decayed learning rate scheduler, preceded by a 6-epoch warm-up phase. Following the warm-up, the learning rate starts at 8e-5 and gradually decays to 0. In our experiments, training is conducted with a total batch size of 512 on 8 NVIDIA A100 80GB GPUs for 6 hours, which early exits at 11-th epoch according to the observation of losses.

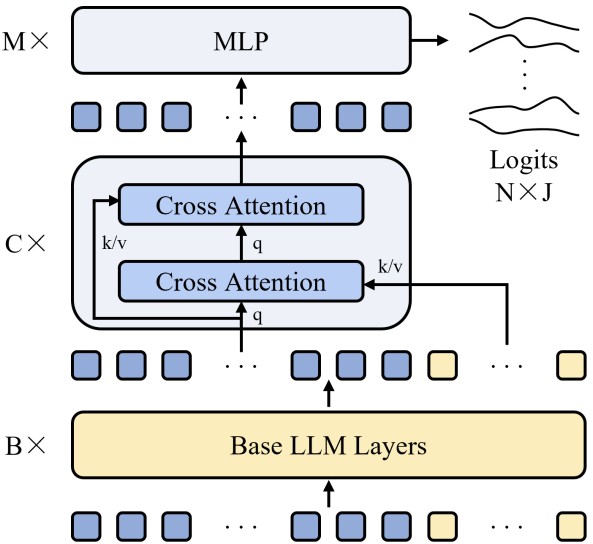

Figure 5: The detailed architecture of TRM

## C ANALYSIS OF TOKEN-RANKING DATASET

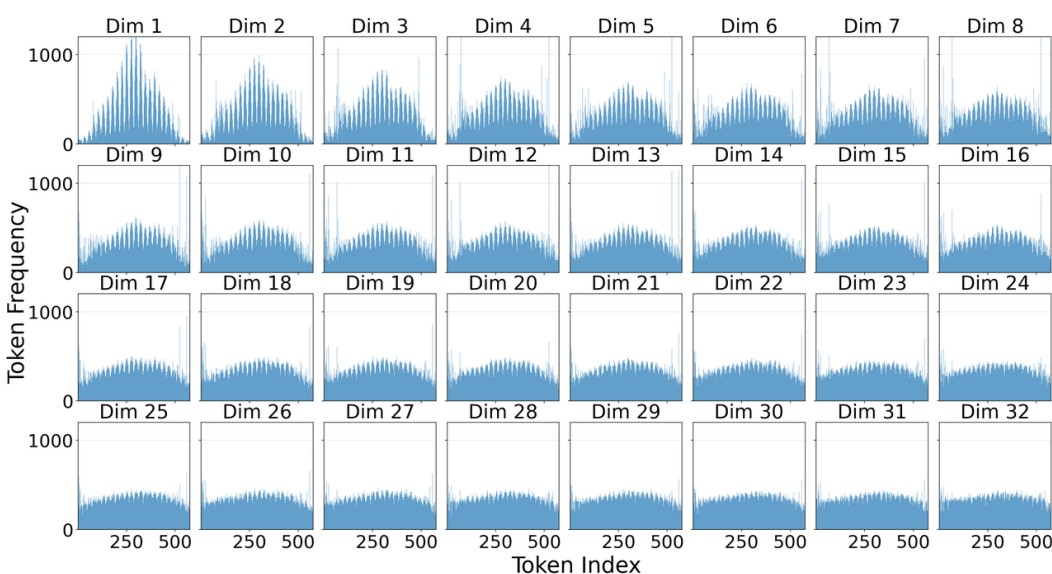

Figure 6: Distribution of tokens-ranking data in training dataset

To explore more characteristics of token-ranking data, we visualize their distributions and make a deep analysis.

In Fig. 6, Fig. 7 and Fig. 8, the 32 sub-plots corresponding to 32 dimensions (i.e., Dim.1 ˜Dim.32) are arranged in descending order of token importance. Each sub-plot takes the original 576 token indices on the x-axis and their frequency in the token-ranking data on the y-axis. In Fig. 9, the x-axis enumerates the Dim1 to Dim32, which are ranked from the highest to the lowest importance. The y-axis records the variance of their frequencies among 576 indices. The means of their frequencies among training data, PoPE and GQA can be calculated through the total frequency divided by the total Dims, which are 353.31, 15.47 and 21.84, respectively.

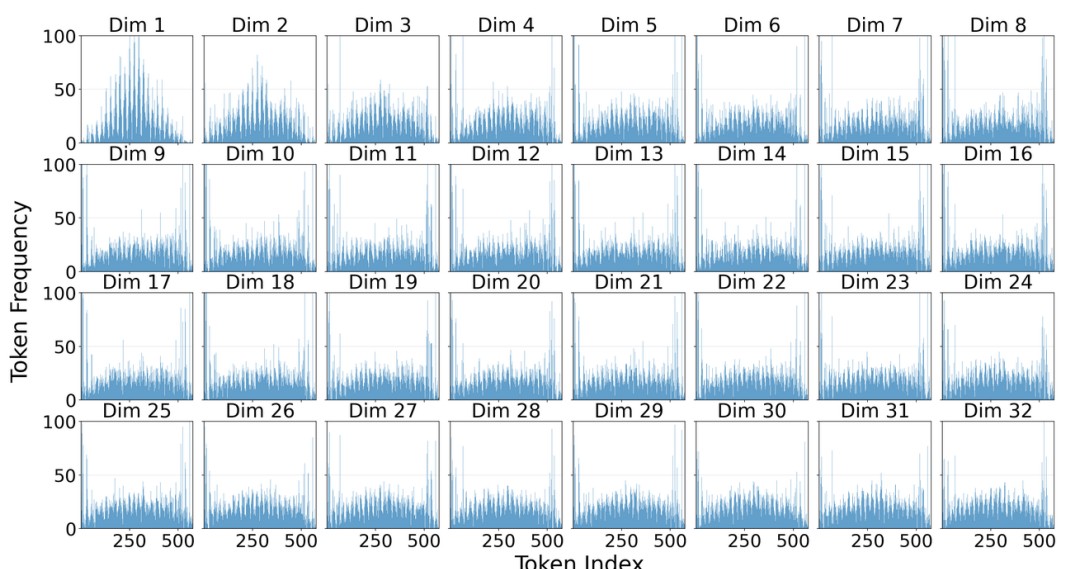

Figure 7: Distribution of token-ranking data in POPE benchmark

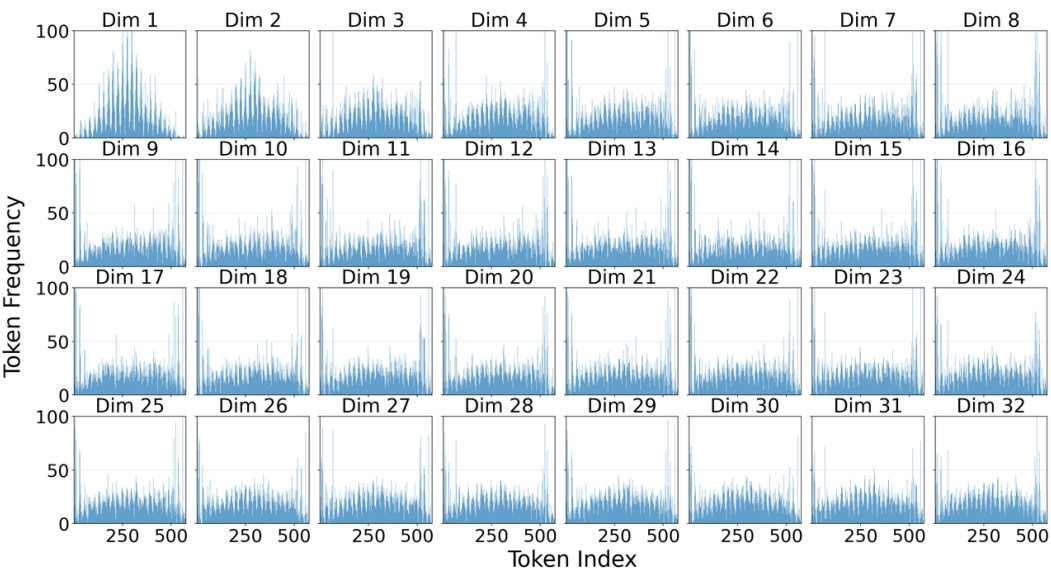

Figure 8: Distribution of token-ranking data in GQA benchmark

These figures jointly reveal that the index distributions in token-ranking data are highly sparse (high-variance) across these benchmarks. The most important token (Dim1 in figures) exhibits maximal sparsity and the narrowest admissible index sets, which means its selection is both predictable and straightforward. By contrast, less important tokens present increasingly diffuse distributions, complicating precise prediction. For example, the variance of the top-3 Dims exhibits a sharp cliff-like drop relative to all subsequent Dims. This progressive dispersion is nearly monotonic of increasing Dim number with decreasing importance in figures.

Therefore, it is concluded that dominant tokens (the earlier serial Dims in token-ranking data) have a large sparsity in indices and a concentration on frequencies, which shows a large variance in Fig.9. In contrast, subordinate tokens have a small sparsity in indices and relatively average distribution on frequencies, which shows a small variance in Fig.9. In conclusion, the large variance on the earlier serial Dims in token-ranking data indicates that important tokens concentrate on a relatively small index set, which are easier to predict. However, the distribution of following Dims in token-

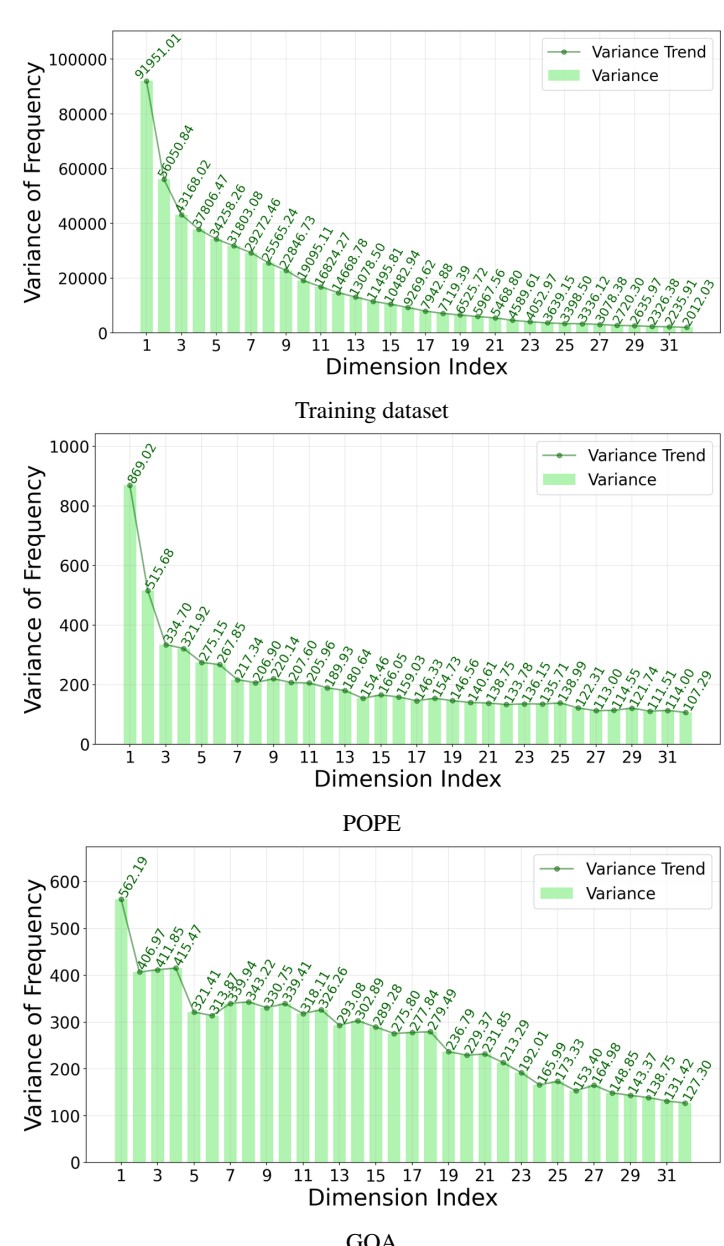

Figure 9: Variance of the frequency among 576 indices in different datasets

ranking data becomes quite uniform, which means these less important tokens are hard to predict. Fortunately, a few important tokens are sufficient for existing tasks in benchmarks. As mentioned in the Sec.4.4, $DTR_{re}$-2 with only top-2 ground-truth of token-ranking data has surpassed the original model Vanilla-576. In this situation, the choice of following tokens may not play an important role on the final performance, thus their distribution of index frequencies becomes uniform.

To further demonstrate the distribution of token-ranking data and make a deep analysis, we make some heatmaps of token-ranking distributions. In the heatmaps, each cell corresponds to an image patch (i.e., a ViT output token) and its color intensity represents the frequencies added among 32 dimensions, which reflects the importance of this patch in token-ranking data.

Fig. 10 reveals that tokens occupying central spatial positions consistently dominate the top ranks of importance, indicating a pronounced bias toward the image center. This phenomenon can be attributed to three mutually reinforcing factors. First, the principal semantic content of natural images

is typically located near the geometric center, and the prevalence of such samples in training data induces a statistical inductive bias that assigns higher information content to central regions. Second, Vision Transformers (ViT) and their variants standardize inputs by padding non-square images to a fixed square canvas before partitioning them into uniform patches. Padding introduces peripheral patches that convey minimal semantic cues, thereby driving the attention distribution inward. Third, the global receptive field of the self-attention mechanism allows central tokens to aggregate gradients from the entire image while simultaneously serving as hubs for information fusion, further amplifying their relative significance.

These findings furnish a theoretically grounded and empirically verifiable prior for token compression. When the importance of individual tokens is ambiguous, the central-bias prior can be leveraged as a heuristic to prioritize centrally located tokens, thus offering a useful strategy for subsequent compression pipelines.

## D ANALYSIS OF TOKEN-RANKING MODEL

As the key component of DTR, Token-ranking Model (TRM) is the output of the offline stage and the actual operating components in the online stage. The performance of TRM directly decides the performance and efficiency of VLM inference with DTR. Though the TRM of DTR achieves a state-of-the-arts performance and acceleration in practice, we still wonder whether it is possible to obtain a more efficient TRM with a better performance.

Therefore, in this section, we first analyze the depth of TRM and show the trade-offs between efficiency and performance. Then we break through the setting (base token-ranking model should keep the same series of LLM used in VLM) in Sec.3.3 and analyze the TRM with an advanced base token-ranking model to further explore the potential of TRM. Finally, we analyze the size of training data to explore the scale-up effect of data for TRM.

### D.1 DEPTH OF TRM

The depth of TRM directly influences the latency overhead. The shallower the TRM depth, the lower the latency overhead. Considering the depth of TRM is mainly decided by the base token-ranking model, to evaluate a more lightweight TRM with shallower depth, we decrease the depth of base token-ranking model from 10 layers in the main text (DTR-L10) to the 3 layers (DTR-L3) in this section. All other parameters keeps same with the implementation in main text.

Tab.4 illustrates the performance across five effective benchmarks with five levels of token compression (i.e., the number of remaining tokens is 32/16/8/4/1). Obviously, on average performance, DTR-L10 achieves the best and DTR-L3 achieves the second best. Both DTR-L10 and DTR-L3 far surpass other comparatives. Generally, the gap between DTR-L10 and DTR-L3 starts from a small value (i.e., 2.9%) and gradually increases as the number of remaining tokens decreases, which suggests the average performance degradation of DTR as the TRM depth and the number of remaining tokens decreases.

It is worth noting that the performance degradation does not exist in all scenarios. DTR-L3 achieves better performance on MME benchmarks with 4 remaining tokens. In fact, the performance gap is quite small on MME benchmarks and GQA, SEED$^{IMG}$, while relatively big on POPE and VQA$^{v2}$. It reveals that the DTR with shallow TRM still works well for simple tasks, while complex task needs deep TRM for better understanding and reasoning.

### D.2 ADVANCED BASE TOKEN-RANKING MODEL

Limited by the minimum size of LLM series in VLM (i.e., at least 7B of Vicuna series in LLaVA) and the necessary depth of base token-ranking model, it is hard to get a more light-weight TRM with a comparable performance. Therefore, we break through the settings (i.e., base token-ranking model should keep the same series of LLM used in VLM) and propose a new method to apply the advanced base token-ranking model instead of the same series of LLM used in VLM. Specifically, we replace the base rank model of Vicuna-7B (the smallest model in Vicuna series) used in the main text with an advanced smaller model Qwen-2.5-1.5B Yang et al. (2024). The following experiments are designed

Table 4: Performance of DTR with different depths

| Method | MME | POPE | VQA$^{v2}$ | GQA | SEED$^{IMG}$ | Avg. |
|---|---|---|---|---|---|---|
| **Vanilla** | 1868 | 86.1 | 78.5 | 62.0 | 66.2 | 100% |
| **Retain 32 Tokens** (↓ 94.4%) | | | | | | |
| **VisPruner** | 1581 | 72.7 | 67.7 | 52.2 | 53.6 | 84.1% |
| **TokenCarve** | 1595 | 69.5 | 69.8 | 53.7 | 56.4 | 85.3% |
| **VisionZip** | 1557 | 68.5 | 67.0 | 51.8 | 53.0 | 82.4% |
| **DTR-L10** | **1679** | **82.5** | **71.4** | **54.6** | **60.2** | **91.1%** |
| **DTR-L3** | 1651 | 79.9 | 68.4 | 52.3 | 58.3 | 88.2% |
| **Retain 16 Tokens** (↓ 97.2%) | | | | | | |
| **VisPruner** | 1347 | 57.0 | 59.6 | 47.0 | 47.6 | 72.4% |
| **TokenCarve** | 1305 | 49.7 | 59.2 | 47.3 | 48.0 | 70.4% |
| **VisionZip** | 1371 | 53.0 | 59.2 | 46.5 | 46.9 | 71.2% |
| **DTR-L10** | **1560** | **77.6** | **67.5** | **51.8** | **57.9** | **86.1%** |
| **DTR-L3** | 1524 | 73.1 | 60.6 | 48.0 | 53.7 | 80.4% |
| **Retain 8 Tokens** (↓ 98.6%) | | | | | | |
| **VisPruner** | 1168 | 38.4 | 50.3 | 42.0 | 41.9 | 60.5% |
| **TokenCarve** | 1035 | 26.8 | 48.2 | 41.4 | 39.5 | 54.9% |
| **VisionZip** | 1084 | 24.2 | 46.9 | 39.9 | 39.4 | 54.0% |
| **DTR-L10** | 1431 | **73.8** | **63.1** | **48.7** | **54.5** | **80.7%** |
| **DTR-L3** | **1466** | 65.0 | 53.7 | 44.4 | 49.5 | 73.8% |
| **Retain 4 Tokens** (↓ 99.3%) | | | | | | |
| **VisPruner** | 1102 | 35.9 | 45.7 | 40.2 | 40.1 | 56.9% |
| **TokenCarve** | 903 | 16.3 | 42.0 | 38.8 | 35.8 | 47.5% |
| **VisionZip** | 926 | 25.2 | 38.9 | 36.6 | 35.3 | 48.1% |
| **DTR-L10** | **1360** | **69.2** | **59.2** | **46.1** | **50.9** | **76.0%** |
| **DTR-L3** | 1339 | 54.0 | 49.1 | 41.9 | 46.2 | 66.9% |
| **Retain 1 Token** (↓ 99.8%) | | | | | | |
| **VisPruner** | 975 | 44.5 | 41.1 | 37.7 | 37.0 | 54.6% |
| **TokenCarve** | – | – | – | – | – | – |
| **VisionZip** | 881 | 26.7 | 38.3 | 36.0 | 34.5 | 47.4% |
| **DTR-L10** | **1167** | **58.9** | **51.0** | **41.2** | **44.3** | **65.8%** |
| **DTR-L3** | 1081 | 45.9 | 41.0 | 39.4 | 41.0 | 57.8% |

to examine whether DTR framework can achieve an efficient VLM inference with different TRMs, thus validating the availability of DTR framework.

Same as the main text, the VLM for acceleration is set as the LLaVA-1.5-7B, the compression layer is set as the input of LLM. To align the latent spaces of Qwen-LLM-1.5B and Vicuna-7B as much as possible, we follow the training pipeline of LLaVA-1.5 series and adopt a two-stage training strategy. In the first stage, we perform one epoch of pretraining on 558K samples, training only an random-initialized projector that connects the LLaVA-1.5-7B's vision encoder and the Qwen-LLM-1.5B to achieve an initial alignment. In the second stage, we fine-tune the Qwen-LLM-1.5B and the projector on 665K samples, where both the MLP and LLM modules are trained to further enhance the alignment. All hyperparameters used during training are consistent with those in LLaVA-1.5 series. Finally, we replace the base ranking model from the first 10 layers of Vicuna-7B with the first 10 layers of this trained Qwen2.5-1.5B. Then we construct the TRM and apply the same token-ranking model training in the main text.

Experimental results are illustrated in Tab.6. Generally, DTR with Qwen (DTR-Qwen) achieves the second best in most scenarios and consistently better performance than challenging comparatives. Specifically, for 32, 16, 8, 4, and 1 remaining vision tokens, DTR-Qwen outperforms challenging comparatives on the POPE dataset at least 10.7%, 32.3%, 77.3%, 61.8%, and 2.0%, respectively, and on the MME dataset over 4.3%, 16.4%, 23.9%, 20.6%, and 8.5%, respectively. It is worth noting that DTR-Qwen even achieves the best performance in some experimental settings (i.e., retaining only 16 and 8 vision tokens on the MME benchmark). Moreover, compared to DTR-L3 with 1.03B base token-ranking model, DTR-Qwen with 0.76B base token-ranking model achieves better performance in almost all scenarios except for retaining 1 token. Therefore, it is clear that DTR still holds potential for further improvement with an advanced base token-ranking model.

Table 5: Performance analysis of Token-Ranking Model under different training data sizes

| Method | MME | POPE | VQA$^{v2}$ | GQA | SEED$^{IMG}$ | Avg. |
|---|---|---|---|---|---|---|
| **Vanilla** | 1868 | 86.1 | 78.5 | 62.0 | 66.2 | 100% |
| **Retain 32 Tokens (↓ 94.4%)** | | | | | | |
| **VisPruner** | 1581 | 72.7 | 67.7 | 52.2 | 53.6 | 84.1% |
| **TokenCarve** | 1595 | 69.5 | **69.8** | **53.7** | 56.4 | 85.3% |
| **VisionZip** | 1557 | 68.5 | 67.0 | 51.8 | 53.0 | 82.4% |
| **DTR-60K** | 1628 | 77.0 | 68.6 | 53.2 | **58.4** | 87.6% |
| **DTR-200K** | **1651** | **79.9** | 68.4 | 52.3 | 58.3 | **88.2%** |
| **Retain 16 Tokens (↓ 97.2%)** | | | | | | |
| **VisPruner** | 1347 | 57.0 | 59.6 | 47.0 | 47.6 | 72.4% |
| **TokenCarve** | 1305 | 49.7 | 59.2 | 47.3 | 48.0 | 70.4% |
| **VisionZip** | 1371 | 53.0 | 59.2 | 46.5 | 46.9 | 71.2% |
| **DTR-60K** | 1512 | 68.1 | **62.6** | **49.5** | **55.3** | **80.6%** |
| **DTR-200K** | **1524** | **73.1** | 60.6 | 48.0 | 53.7 | 80.4% |
| **Retain 8 Tokens (↓ 98.6%)** | | | | | | |
| **VisPruner** | 1168 | 38.4 | 50.3 | 42.0 | 41.9 | 60.5% |
| **TokenCarve** | 1035 | 26.8 | 48.2 | 41.4 | 39.5 | 54.9% |
| **VisionZip** | 1084 | 24.2 | 46.9 | 39.9 | 39.4 | 54.0% |
| **DTR-60K** | 1416 | 59.6 | **56.9** | **46.3** | **51.6** | **74.0%** |
| **DTR-200K** | **1466** | **65.0** | 53.7 | 44.4 | 49.5 | 73.8% |
| **Retain 4 Tokens (↓ 99.3%)** | | | | | | |
| **VisPruner** | 1102 | 35.9 | 45.7 | 40.2 | 40.1 | 56.9% |
| **TokenCarve** | 903 | 16.3 | 42.0 | 38.8 | 35.8 | 47.5% |
| **VisionZip** | 926 | 25.2 | 38.9 | 36.6 | 35.3 | 48.1% |
| **DTR-60K** | 1277 | 49.5 | **52.3** | **43.7** | **48.0** | **67.1%** |
| **DTR-200K** | **1339** | **54.0** | 49.1 | 41.9 | 46.2 | 66.9% |
| **Retain 1 Token (↓ 99.8%)** | | | | | | |
| **VisPruner** | 975 | 44.5 | **41.1** | 37.7 | 37.0 | 54.6% |
| **TokenCarve** | – | – | – | – | – | – |
| **VisionZip** | 881 | 26.7 | 38.3 | 36.0 | 34.5 | 47.4% |
| **DTR-60K** | **1082** | 42.7 | 41.0 | **39.9** | **41.7** | 57.4% |
| **DTR-200K** | 1081 | **45.9** | 41.0 | 39.4 | 41.0 | **57.8%** |

## D.3 SCALE OF TRAINING DATASET

To explore the scale-up effect of data and investigate whether increasing the training dataset can offset the performance degradation, we conducted additional experiments using DTR-L3 on two scales of token-ranking dataset for training. The first contains over 200K samples mentioned in the main text. The second contains 60K samples randomly sampled from 200K samples in this section.

Tab.5 illustrates the comparison results. Generally, DTR-60k and DTR-200K achieve the best or the second best in all scenarios and consistently surpass challenging comparatives. Specifically, there only exists a small gap between DTR-60k and DTR-200K, which is only up to 0.60% and with an average 0.32% on average performance. Surprisingly, DTR trained on the larger 200K dataset ( DTR-200K) does not consistently outperform its 60K counterpart (DTR-60K). In fact, DTR-60K is the best when the number of remaining tokens is set to 16, 8, or 4. This phenomenon indicates that the data maybe somewhat excessive for the current TRM. Therefore, it is suggested that we can improve the learning ability of TRM to make full use of the training data and achieve a better performance, or delicately filter the training data and get a promising DTR with much less training overhead.

## E QUALITATIVE ANALYSIS OF DTR

In this section, we list some examples in our experiments to make a qualitative analysis of DTR. More qualitative analyses will be summarized in the upcoming release of DTR project.

Table 6: DTR with advanced base toke-ranking model.

| Method | POPE | MME | Avg. |
|---|---|---|---|
| **Retain 32 Tokens** (↓ 94.4%) | | | |
| VisPruner | 72.7 | 1581 | 84.5% |
| TokenCarve | 69.5 | 1595 | 83.1% |
| VisionZip | 68.5 | 1557 | 81.5% |
| DTR-L3 | 79.9 | 1651 | 90.6% |
| DTR-L10 | **82.5** | **1679** | **92.9%** |
| DTR-Qwen | 80.5 | 1663 | 91.3% |
| **Retain 16 Tokens** (↓ 97.2%) | | | |
| VisPruner | 57.0 | 1347 | 69.2% |
| TokenCarve | 49.7 | 1305 | 63.8% |
| VisionZip | 53.0 | 1371 | 67.5% |
| DTR-L3 | 73.1 | 1524 | 83.2% |
| DTR-L10 | **77.6** | 1560 | **86.8%** |
| DTR-Qwen | 75.4 | **1596** | 86.5% |
| **Retain 8 Tokens** (↓ 98.6%) | | | |
| VisPruner | 38.4 | 1168 | 53.6% |
| TokenCarve | 26.8 | 1035 | 43.3% |
| VisionZip | 24.2 | 1084 | 43.1% |
| DTR-L3 | 65.0 | **1466** | 77.0% |
| DTR-L10 | **73.8** | 1431 | **81.2%** |
| DTR-Qwen | 68.1 | 1447 | 78.3% |
| **Retain 4 Tokens** (↓ 99.3%) | | | |
| VisPruner | 35.9 | 1102 | 50.3% |
| TokenCarve | 16.3 | 903 | 33.7% |
| VisionZip | 25.2 | 926 | 39.4% |
| DTR-L3 | 54.0 | 1339 | 67.2% |
| DTR-L10 | **69.2** | **1360** | **76.6%** |
| DTR-Qwen | 58.1 | 1329 | 69.3% |
| **Retain 1 Token** (↓ 99.8%) | | | |
| VisPruner | 44.5 | 975 | 51.9% |
| TokenCarve | - | - | - |
| VisionZip | 26.7 | 881 | 39.1% |
| DTR-L3 | 45.9 | 1081 | 55.6% |
| DTR-L10 | **58.9** | **1167** | **65.4%** |
| DTR-Qwen | 45.4 | 1058 | 54.7% |

Fig. 11 illustrates the qualitative performance of DTR across different remaining token numbers. Notably, DTR is capable of producing accurate results even with just one image token. Fig. 12 illustrates a comparison of DTR, VisionZip, and TokenCarve under the constraint of 1 image token. In this scenario, VisionZip and TokenCarve struggle to generate correct answers, whereas DTR maintains its accuracy. This highlights an obvious limitation of current model-based methods, which lack a fine-grained understanding when compression with extremely low remaining token numbers.

Fig.13 illustrates that reducing the number of remaining vision tokens also leads to a performance degradation in DTR due to the information capacity and spatial location of tokens. The essence of DTR based on the data-driven method is still a form of probabilistic learning, which naturally achieves better performance with more samplings to cover the necessary tokens. Also worth noting, not only main objects but also tiny objects require large remaining tokens. For example, the umbrella exhibits a more compact structure than the teddy bear, thus requiring a larger number of vision tokens to cover the necessary information. This observation indicates that a deeper semantic understanding of tokens is still required to select necessary tokens, which is hard to achieve through model-driven methods. Therefore, more efforts will be made on data-driven methods in the future.

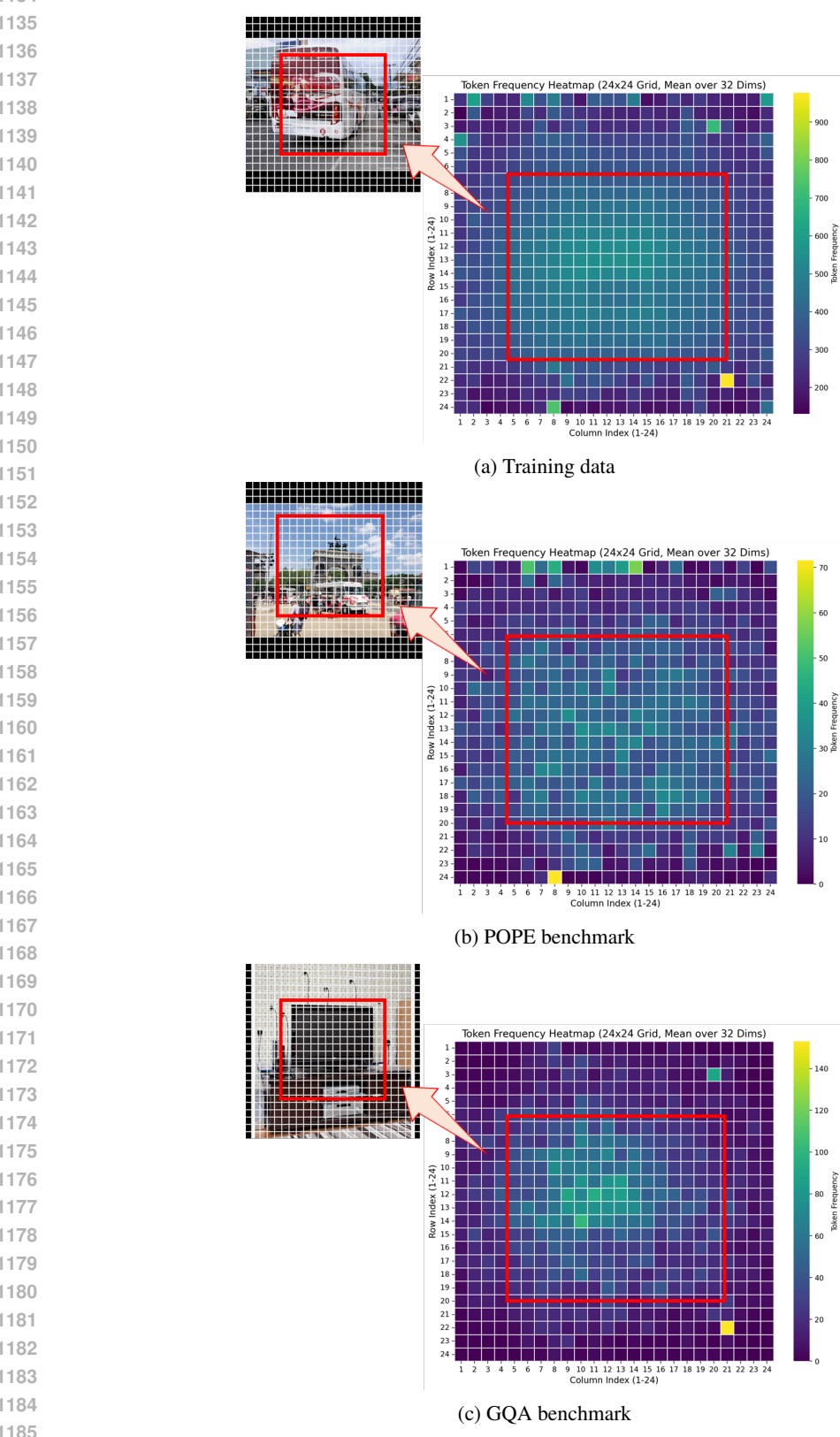

(a) Training data

(b) POPE benchmark

(c) GQA benchmark

Figure 10: Heatmap of remaining-token frequencies.

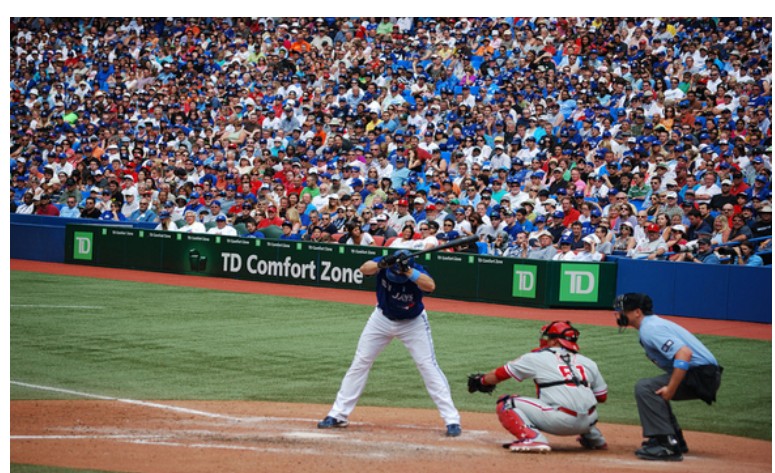

*Question: Is the catcher playing or waiting?*

*Answer: Waiting*

*Question: What are the ornaments made of glass hanging from?*

*Answer: Tree*

Figure 11: DTRs with 32, 16, 8, 4, and 1 token achieve the same correct answer.

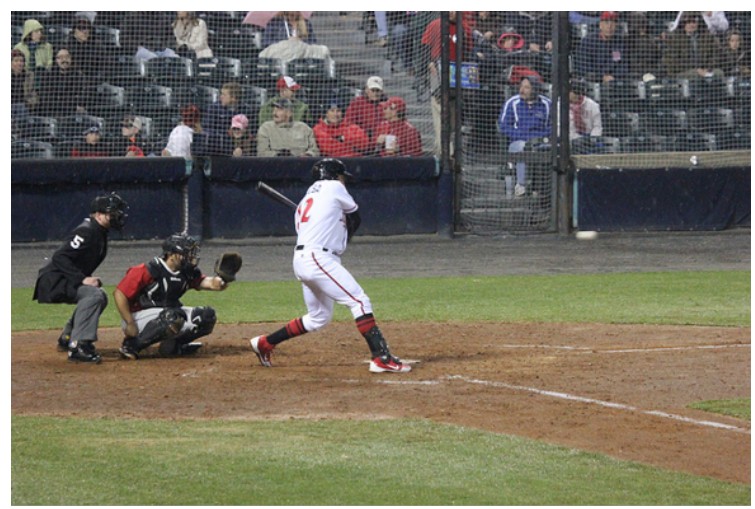

*Question: How big is the leather glove?*

DTR: large    VisionZip: small    TokenCarve: small    *Answer: large*

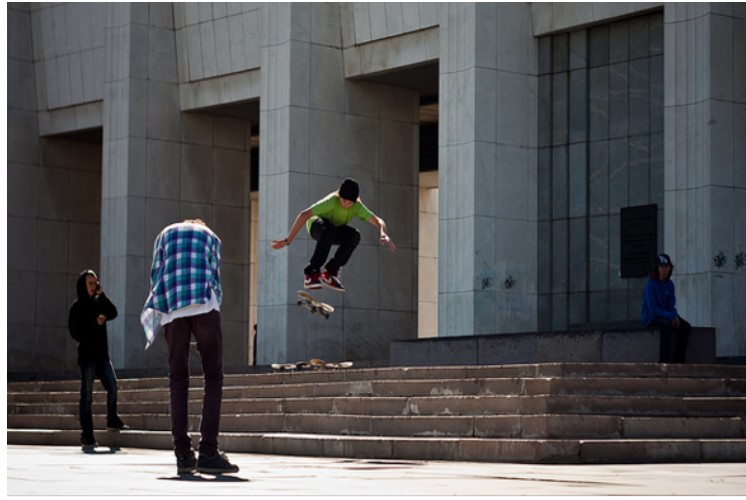

*Question: Does the person that is bending look male?*

DTR: yes    VisionZip: no    TokenCarve: no    *Answer: yes*

Figure 12: Qualitative results of DTR, VisionZip, and TokenCarve under 1 token.

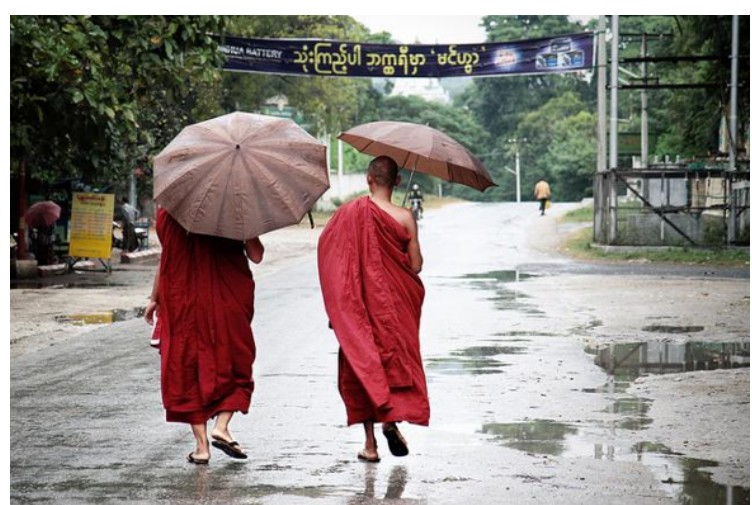

*Question: What does the chubby man hold?*

*Answers: Umbrella.*

DTR-32 DTR-16 DTR-8 DTR-4 DTR-1

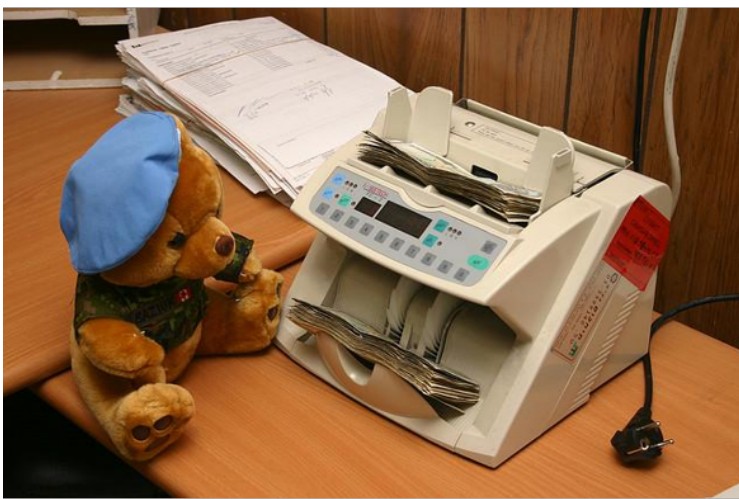

*Question: Which kind of toy is soft?*

*Answers: Stuffed bear.*

DTR-32 DTR-16 DTR-8 DTR-4 DTR-1

Figure 13: The comparison among DTR with different numbers of remaining tokens.

