# OpenReview forum: "DTR: Towards optimal token compression with data-driven token ranking for efficient visual-language model inference"
_ICLR.cc/2026/Conference — Submitted to ICLR 2026_

### Official Review · Reviewer_haQJ · 2025-10-17

**Soundness:** 3
**Presentation:** 3
**Contribution:** 3
**Rating:** 4
**Confidence:** 4

**Summary:**

The paper addresses the computational inefficiency of vision-language models caused by excessive visual tokens during inference. It proposes DTR, a data-driven token ranking framework that replaces handcrafted model-driven compression heuristics with a learned token importance predictor. The core contributions include: (1) a method to construct token-ranking datasets using greedy search over token subsets, (2) a token-ranking model (TRM) trained to predict token importance rankings, and (3) a plug-and-play integration scheme for runtime token filtering.

**Strengths:**

1. The method is widely evaluated on 8 diverse benchmarks against 8 strong baselines.

2. The proposed TRM is plug-and-play, requiring no architectural changes to VLMs.

3. The ablation study is abundant and comprehensive.

**Weaknesses:**

1. The paper is not well-written and hard to read, with several typos that I cannot understand. For example, the citation of the paper all comes with author(year) but does not mention the name of the paper, which is weird.  Also, line371 does not contain any information. What do you mean by adding that? I suggest that the author comprehensively revise the paper including but not limited to the typos noted above.

2. The results are not exciting. First, the results when retaining 64 tokens are substantially inferior to other methods, which challenges the utility of the method. Also, although improvements are achieved when the number of tokens becomes fewer, most comparison methods are uniform or random, which is not exciting. More methods are supposed to be incorporated to validate the effectiveness.

3. As you mention in line465, the overhead of DTR is a crucial question of the method, which may offset gains for small batches or simple images. Based on the results, we are unable to know whether the performance gain comes from additional computation or the effectiveness of the method. I recommend the author explain more about this, including but not limited to the actual latency, comparing under the same flops other than the same token and so on.

4. The paper claims the "global optimum token compression", but it requires training without analytical support beyond empirical results. I would like to see more theoretical proof of how your token compression is optimal but no discussion is shown about this in the paper, which significantly reduces the persuasiveness of the article. Also, the choice of greedy search over optimal combinatorial search lacks theoretical guarantees on ranking quality.

5. ​​Sparse analysis of multimodal interactions​​: The role of text tokens in guiding visual token ranking is underexplored.

**Questions:**

1. The greedy algorithm for forward passes are computational expensive, especially for large N. How do you deal with it and are there any quantitative results?

2. Is the TRM latency overhead amortizable across batches, and what are optimal batch sizes for real-world deployment?

3. Given the high computational consumption, what is the scalability of the proposed method on larger models, like 32B, different architectures, like Qwen?

4. The diversity of evaluated datasets is somewhat narrow and more results are expected to be conducted to validate this. For example, how does DTR perform on tasks requiring fine-grained spatial reasoning (e.g., object counting), long context understanding and could task-specific ranking models help?

5. DTR relies heavily on the quality of ranking. Are there failure modes where DTR's rankings degrade VLM performance (e.g., adversarial images), and how can robustness be improved?

---

> ### Author Response · Authors · 2025-12-02
> **Response to Weakness1,2,3**
>
> Weaknesses1: The paper is not well-written and hard to read, with several typos that I cannot understand. For example, the citation of the paper all comes with author(year) but does not mention the name of the paper, which is weird. Also, line371 does not contain any information. What do you mean by adding that? I suggest that the author comprehensively revise the paper including but not limited to the typos noted above.
>
> Response to Weakness1:
>
> Thank the reviewer for this question. We have improved the writing and made the manuscript easier to read.
> Specifically, for "the citation of the paper all comes with author(year) but does not mention the name of the paper", the name of the work is added in the citation. Because of the template of this conference, the citation only shows the author(year) and has a hyperlink for jumping to the reference. We are sorry for the inconvenience to the reviewer and have made citations as clear as possible.
>
> Then, for  "line371 does not contain any information" (i.e.  the results of TokenCarve in Table1 in the manuscript), the reason is added in 375 (i.e., as a note below the Table), which is "*The project of TokenCarve can not be implemented for retaining 1 token, so there exist no results".
>
> Weaknesses2: The results are not exciting. First, the results when retaining 64 tokens are substantially inferior to other methods, which challenges the utility of the method. Also, although improvements are achieved when the number of tokens becomes fewer, most comparison methods are uniform or random, which is not exciting. More methods are supposed to be incorporated to validate the effectiveness.
>
> Response to Weakness2:
>
> Thank the reviewer for this question. Due to the state-of-the-art performance of DTR with 32 remaining tokens and the limitation of time, we indeed did not train DTR with 64 remaining tokens.
>
> According to the main results demonstrated in table.1 in our manuscript, DTR achieves the consistently best performance among 6 comparatives when retaining tokens≤32. As the remaining tokens decrease, the superiority of DTR becomes even more pronounced.
> Moreover, according to the Table.1 in the manuscript, it is exciting that DTR with 32 remaining tokens even surpasses the comparatives with 64 remaining tokens except for TokenCarve. In fact, DTR with 32 remaining tokens can also surpass the TokenCarve in 3/8 benchmarks (i.e., POPE, SQA^{IMG} and VizWiz).
>
> Weaknesses3: As you mention in line465, the overhead of DTR is a crucial question of the method, which may offset gains for small batches or simple images. Based on the results, we are unable to know whether the performance gain comes from additional computation or the effectiveness of the method. I recommend the author explain more about this, including but not limited to the actual latency, comparing under the same flops other than the same token and so on
>
> Response to Weaknesses3:
>
> Thank the reviewer for this valuable question. When the budget of parallel computation is sufficient, all token compression methods indeed have no significant effect, because the main bottleneck of latency comes from the serial computation corresponding to the depth of VLMs (i.e., the number of layers), rather than the parallel computation corresponding to the width of VLMs (i.e. the number of tokens). In this case, TRM brings more serial computation and latency. However,  in practice, the token compression methods mainly work for the large-scale inference scenarios, where exist a significant parallel computing bottleneck. Then DTR achieves a significant inference acceleration according to the table.3 in the manuscript (decrease the prefilling latency per sample from over 100ms to over 10ms, the decoding latency per word from over 3ms to over 1ms.
>
> Moreover, for a clear demonstration of the overheads and gains of DTR, Table.3-1 illustrates the end-to-end consumption of latency and memory between vanilla VLM (i.e., vanilla-576) and VLM with 32 remaining tokens through DTR (i.e. DTR-32 ).
>
> Table.3-1 The end-to-end consumption latency
>
> | BATCH SIZE | vanilla-576(ms) | DTR-32(ms) | vanilla-576(MB) | DTR-32(MB) |
> |------------|-----------------|------------|-----------------|------------|
> | 1          | 182.6           | 104.5      | 16882           | 17639      |
> | 4          | 148             | 63.7       | 19652           | 19293      |
> | 8          | 139.6           | 54.6       | 23662           | 21347      |
> | 16         | 133.9           | 51.8       | 30248           | 26489      |
> | 32         | 132.1           | 50.9       | 44210           | 36133      |
>
> Obviously, DTR achieves consistent superiority in latency. However, due to the overhead of TRM model, DTR consumes more memory when batch size=1. This phenomenon changes when batch size=4. As the batch size increases, DTR saves more and more memory.

---

> ### Author Response · Authors · 2025-12-02
> **Response to Weakness 4,5**
>
> Weaknesses4: The paper claims the "global optimum token compression", but it requires training without analytical support beyond empirical results. I would like to see more theoretical proof of how your token compression is optimal but no discussion is shown about this in the paper, which significantly reduces the persuasiveness of the article. Also, the choice of greedy search over optimal combinatorial search lacks theoretical guarantees on ranking quality.
>
> Response to Weaknesses 4:
>
> Thank the reviewer for this insightful question. As mentioned in the title of this paper "TOWARDS OPTIMAL TOKEN COMPRESSION
> WITH DATA-DRIVEN TOKEN RANKING", DTR is designed for approaching the optimal token compression. However, there still exists a noticeable distance away from the optimal token compression, due to the non-optimal greedy search algorithm and insufficient modeling or training.
>
> Although it cannot achieve optimality, DTR points the way toward the optimal token compression through data-driven paradigm. Specifically, TRM achieves optimality if the following three assumptions are satisfied:
>
> Assumption 1: TRM can be an arbitrary function approximator. This assumption can be satisfied in theory according to the universal approximation theorem[1], which proves that  neural networks with depth>3 can be universal approximators for any function. Obviously, the TRM model is deeper than 3, hence the assumption can be satisfied.
>
> Assumption 2: The labels of the training data are global optimum. This assumption can be satisfied in theory through traversal labeling. In practice, it's not worth traversing the set for the global optimum, because solutions close to the global optimum are already good enough. For example, as demonstrated in Table.1 in our manuscript, the upper-bound of DTR through the greedy algorithm (i.e. DTR_{ub}-32 ) can even yield about a +29% relative improvement than the VLM without compression.
>
> Assumption 3: The training data and test data are independent and identically distributed (i.i.d.). This assumption is adopted in most large model works, considering the tremendous amount of widely-gathered training data for large models in practice.
>
> [1] Cybenko G. Approximation by superpositions of a sigmoidal function[J]. Mathematics of control, signals and systems, 1989, 2(4): 303-314.
>
> Weaknesses5: Sparse analysis of multimodal interactions: The role of text tokens in guiding visual token ranking is underexplored.
>
> Thank the reviewer for this valuable question. The multimodal interactions lie in the TRM, which shows TRM ranks the image tokens based on the guiding of text tokens.
>
> Response to Weakness 5:
>
> Generally, first, TRM lets text tokens merge the image tokens, which helps text token adjust its semantics according to the contents of images and the instructions from text. Then TRM lets the image tokens merge the text tokens, which means the image tokens is judged by the full semantics from text tokens merged with image tokens.
> Specifically, according to the architecture of TRM (i.e., Figure 5 in our manuscript), first, the text token merges the images token at the base LLM layers, then, the image tokens evaluate itself according to the text tokens through cross attention layers.

---

> ### Author Response · Authors · 2025-12-02
> **Response to Question 1,2,3**
>
> Question1：The greedy algorithm for forward passes are computational expensive, especially for large N. How do you deal with it and are there any quantitative results?
>
> Response to Question1:
>
> Thank the reviewer for this valuable question. As mentioned in Section 3.2 in the manuscript, the greedy algorithm decreases the complexity from $O(N!)$ to $O(N^2)$ for ranking original N tokens.  However, in practice, a much smaller number M of remaining tokens for ranking can achieve the nearly same performance as the original token number N.
>
> For example, for the N=576 in LLaVA-1.5 models, the M=32 of DTR achieves the 94% performance of the original model with N=576 tokens. Therefore, even for a much larger N,  the number of remaining tokens for ranking M can be set as a relatively fixed number 32 or 64. Then the complexity further decreases to the $O(N)$, which is affordable for even larger N.
>
> Moreover, the greedy algorithm is updated to a Bayesian optimization algorithm in our future work, which can achieve the same performance with a relatively much smaller searching budget.
>
> Question2: Is the TRM latency overhead amortizable across batches, and what are optimal batch sizes for real-world deployment?
>
> Response to Question 2:
>
> Thank the reviewer for this insightful question. The latency of TRM as well as the VLM can be amortizable across batches. In practice, the inference engine usually employs as large batch size as possible to make full use of the computation resource. Considering the dynamic computation resource and task load in the real-world deployment, the optimal batch size and control scheme are also dynamic. More research will be done in the near future for this valuable question.
>
> Question3: Given the high computational consumption, what is the scalability of the proposed method on larger models, like 32B, different architectures, like Qwen?
>
> Response to Question 3:
>
> Thank the reviewer for this valuable question. Because DTR trains an individual token ranking model to learn the underlying function of self-gathered token ranking data, the differences of VLMs have no impact on DTR.
>
> As mentioned in 154, DTR is a framework for any VLM rather than a model for a specific VLM. The pipeline of DTR requires no additional capabilities of VLM but only the inference with different tokens of a given VLM, which is feasible for all open-source VLMs. Through the pipeline, DTR can build a TRM for any open-source VLM.
> To demonstrate the generalization ability, DTR on LLaVA-13B are demonstrated in the following table, DTR on more VLMs will be released in the following github project. Obviously, DTR achieves a better performance on LLaVA-13B than LLaVA-7B.
>
> Table.3-2 DTR on LLaVA-7B and LLaVA-13B.
> | Method         | MME                       | POPE  | SEED-IMG | GQA   | Avg.    |
> |----------------|---------------------------|-------|----------|-------|---------|
> |                              Retain 576 Tokens (100%)                                   |
> | Vanilla7B-576  | 1868                      | 86.1  | 66.2     | 62.0  | 100.00% |
> | Vanilla13B-576 | 1819                      | 86.1  | 68.2     | 63.3  | 100.00% |
> |                              Retain 32 Tokens (↓94.4%)                                   |
> | DTR-LLaVA7B    | 1679                      | 82.5  | 60.2     | 54.6  | 90.11%  |
> | DTR-LLaVA13B   | 1751                      | 84.4  | 64.4     | 59.7  | 96.25%  |
> |Retain 16 Tokens (↓94.4%)                                                               |
> | DTR-LLaVA7B    | 1560                      | 77.6  | 57.9     | 51.8  | 83.91%  |
> | DTR-LLaVA13B   | 1683                      | 81.7  | 63.3     | 58.7  | 92.65%  |
> |                              Retain 8 Tokens (↓98.6%)                                   |
> | DTR-LLaVA7B    | 1431                      | 73.8  | 54.5     | 48.7  | 77.22%  |
> | DTR-LLaVA13B   | 1600                      | 78.5  | 61.2     | 56.8  | 88.25%  |
> |                              Retain 4 Tokens (↓99.3%)                                   |
> | DTR-LLaVA7B    | 1360                      | 69.2  | 50.9     | 46.1  | 73.29%  |
> | DTR-LLaVA13B   | 1510                      | 73.5  | 58.1     | 54.2  | 83.29%  |
> |                              Retain 1 Tokens (↓99.8%)                                   |
> | DTR-LLaVA7B    | 1167                      | 58.9  | 44.3     | 41.2  | 62.98%  |
> | DTR-LLaVA13B   | 1259                      | 66.1  | 51.4     | 48.6  | 69.99%  |
>
> Considering the LLAVA-1.5-7B is the most widely-used model in existing works and has the most comprehensive experimental results for comparison,  it is selected as the main model to evaluate various methods. More experimental results on various models will be added in the appendix and the following github project.

---

> ### Author Response · Authors · 2025-12-02
> **Response to Question 4,5**
>
> Question4: The diversity of evaluated datasets is somewhat narrow and more results are expected to be conducted to validate this. For example, how does DTR perform on tasks requiring fine-grained spatial reasoning (e.g., object counting), long context understanding and could task-specific ranking models help?
>
> Response to Question4:
> Thank the reviewer for this valuable question. We find there exist some spatial reasoning and long context understanding in GQA benchmarks. Through fine-tuning on the GQA benchmarks, DTR-FineTune achieves a significant improvement to DTR. Some examples of the question in GQA and the fine-tuning results are demonstrated in the following table.
>
> Table.3-3 Examples and Fine-tuning results of GQA.
>
> |Examples for the combination of spatial location and detection|
> |--------------------|
> |Is the curtain to the left of a pillow?|
> |Does the building behind the car look brown?|
>
> |Examples for large-scale understanding|
> |--------------------|
> |What is on the table made of wood?|
> |Which kind of vegetable is on the cutting board|
>
> | Method             | GQA   |
> |--------------------|-------|
> | LLaVA-576                | 61.97 |
> | Retain 32 tokens                   |
> | DTR                         | 54.11 |
> | DTR-FineTune            | 62.23 |
> | Retain 16 tokens                   |
> | DTR                         | 51.16 |
> | DTR-FineTune            | 60.05 |
>
> Here, the GQA benchmark is divided into 8000 training samples and 4578 test samples. It is worth noting that the original VLM (i.e., LLaVA-576) keeps the nearly same performance (i.e, the difference < 0.1) between the full benchmarks and divided benchmarks, which shows the division makes little difference on the testing.
>
> Question5: DTR relies heavily on the quality of ranking. Are there failure modes where DTR's rankings degrade VLM performance (e.g., adversarial images), and how can robustness be improved?
>
> Response to Question5:
>
> Thank the reviewer for this insightful question. There exist some failure cases of DTR due to insufficient modeling and training. However, through the analysis of top-100 failure cases in the training dataset, we can not find a certain mode of failure case. Through a comprehensive investigation on the 8 benchmarks, the following pattern emerges:  on samples where the original VLM performs poorly, DTR tends to perform even worse—a phenomenon that also aligns with intuition of model compression.

---

### Official Review · Reviewer_TCpG · 2025-10-28

**Soundness:** 3
**Presentation:** 3
**Contribution:** 3
**Rating:** 4
**Confidence:** 4

**Summary:**

This paper proposed Data-driven Token Ranking (DTR): a plug-and-play ranking model trained on automatically collected token-importance orders from standard VLM datasets. At runtime, DTR predicts a ranked list from the input vision–text tokens and filters to a user-specified budget for acceleration. Across 8 mainstream benchmarks, DTR delivers state-of-the-art compression, and analysis indicates substantial headroom—often matching or surpassing vanilla VLMs with far fewer tokens.

**Strengths:**

1. Using a novel two-stage algorithm,  offline and online; offline: the end-to-end loss for ranking the selected token lists to create a token ranking dataset and train a TRM for automatically selecting the top related tokens; online: inference with the plug-and-play TRM, combined with a user-defined number of tokens.

2. The upper bound of the method is surprisingly achieved SOTA in a really high pruning ratio of the vision tokens.

3. The paper is very well written and easy to read, with a clear logical flow.

**Weaknesses:**

1. Generalization yet to be verified: The paper lacks experiments on different models and numbers of parameters; they only conduct experiments on the LLaVA-7B; effectiveness on other architectures (e.g., LLaVA-OV[1], InstructBLIP[2], Qwen-VL[3]) and other numbers of parameters(e.g., LLaVA-13B) remains to be validated.

2. Baseline selection is not accurate: The comparison with existing methods is not entirely fair, as some baselines are not aligned in settings or optimization conditions. For example, the baselines are all training-free methods, which are substantially different from the training setting in this paper. Therefore, more methods should be compared, such as PDrop[4], M3[5], FastVLM[6], and so on.

3. Compare to other SOTA baselines: I found another SOTA baseline, QueCC[7], that also claims they select a very minimum vision token and still gain great accuracy.

[1] Bo Li, Yuanhan Zhang, Dong Guo, Renrui Zhang, Feng Li, Hao Zhang, Kaichen Zhang, Yanwei Li, Ziwei Liu, and Chunyuan Li. Llava-onevision: Easy visual task transfer. ArXiv, 2024a.

[2] Wenliang Dai and Junnan Li and Dongxu Li and Anthony Meng Huat Tiong and Junqi Zhao and Weisheng Wang and Boyang Li and Pascale Fung and Steven Hoi. InstructBLIP: Towards General-purpose Vision-Language Models with Instruction Tuning. ArXiv, 2023a.

[3] Jinze Bai, Shuai Bai, Shusheng Yang, Shijie Wang, Sinan Tan, Peng Wang, Junyang Lin, Chang Zhou, and Jingren Zhou. Qwen-vl: A versatile vision-language model for understanding, localization, text reading, and beyond. arXiv preprint arXiv:2308.12966, 2023.

[4] Long Xing, Qidong Huang, Xiaoyi Dong, Jiajie Lu, Pan Zhang, Yuhang Zang, Yuhang Cao, Conghui He, Jiaqi Wang, Feng Wu, et al. Pyramiddrop: Accelerating your large vision-language models via pyramid visual redundancy reduction. CVPR, 2025.

[5] Cai, Mu and Yang, Jianwei and Gao, Jianfeng and Lee, Yong Jae. M3: Matryoshka Multimodal Models. ICLR, 2025.

[6] Pavan Kumar Anasosalu Vasu, Fartash Faghri, Chun-Liang Li, Cem Koc, Nate True, Albert Antony, Gokul Santhanam, James Gabriel, Peter Grasch, Oncel Tuzel, Hadi Pouransari. FastVLM: Efficient Vision Encoding for Vision Language Models. CVPR, 2025.

[7] Li, K. Y., Goyal, S., Semedo, J. D., & Kolter, J. Z. Inference Optimal VLMs Need Fewer Visual Tokens and More Parameters. ICLR, 2025.

**Questions:**

1. Upper bound–TRM gap at 32 tokens. The paper reports that the "upper bound” yields about a +29% relative improvement, whereas the learned TRM preserves about 94% of the baseline at 32 tokens. Could you diagnose the sources of this gap? Is it caused by an insufficient modeling or training of the TRM?

I will consider raising my score if all my concerns are solved.

---

> ### Author Response · Authors · 2025-12-02
> **Response to Weakness1**
>
> Weakness1：Generalization yet to be verified: The paper lacks experiments on different models and numbers of parameters; they only conduct experiments on the LLaVA-7B; effectiveness on other architectures (e.g., LLaVA-OV[1], InstructBLIP[2], Qwen-VL[3]) and other numbers of parameters(e.g., LLaVA-13B) remains to be validated.
>
> Response to Weakness1:
> Thank the reviewer for this valuable question. Because DTR trains an individual token ranking model to learn the underlying function of self-gathered token ranking data, the differences of VLMs have no impact on DTR.
>
> As mentioned in 154, DTR is a framework for any VLM rather than a model for a specific VLM. The pipeline of DTR requires no additional capabilities of VLM but only the inference with different tokens of a given VLM, which is feasible for all open-source VLMs. Through the pipeline, DTR can build a TRM for any open-source  VLM.
>
> To demonstrate the generalization ability, DTR on LLaVA-13B are demonstrated in the following table, DTR on more VLMs will be released in the following github project. Obviously, DTR achieves a better performance on LLaVA-13B than LLaVA-7B.
> Table.2-1 DTR on LLaVA-7B and LLaVA-13B.
> | Method         | MME                       | POPE  | SEED-IMG | GQA   | Avg.    |
> |----------------|---------------------------|-------|----------|-------|---------|
> |                              Retain 576 Tokens (100%)                                   |
> | Vanilla7B-576  | 1868                      | 86.1  | 66.2     | 62.0  | 100.00% |
> | Vanilla13B-576 | 1819                      | 86.1  | 68.2     | 63.3  | 100.00% |
> |                              Retain 32 Tokens (↓94.4%)                                   |
> | DTR-LLaVA7B    | 1679                      | 82.5  | 60.2     | 54.6  | 90.11%  |
> | DTR-LLaVA13B   | 1751                      | 84.4  | 64.4     | 59.7  | 96.25%  |
> |Retain 16 Tokens (↓94.4%)                                                               |
> | DTR-LLaVA7B    | 1560                      | 77.6  | 57.9     | 51.8  | 83.91%  |
> | DTR-LLaVA13B   | 1683                      | 81.7  | 63.3     | 58.7  | 92.65%  |
> |                              Retain 8 Tokens (↓98.6%)                                   |
> | DTR-LLaVA7B    | 1431                      | 73.8  | 54.5     | 48.7  | 77.22%  |
> | DTR-LLaVA13B   | 1600                      | 78.5  | 61.2     | 56.8  | 88.25%  |
> |                              Retain 4 Tokens (↓99.3%)                                   |
> | DTR-LLaVA7B    | 1360                      | 69.2  | 50.9     | 46.1  | 73.29%  |
> | DTR-LLaVA13B   | 1510                      | 73.5  | 58.1     | 54.2  | 83.29%  |
> |                              Retain 1 Tokens (↓99.8%)                                   |
> | DTR-LLaVA7B    | 1167                      | 58.9  | 44.3     | 41.2  | 62.98%  |
> | DTR-LLaVA13B   | 1259                      | 66.1  | 51.4     | 48.6  | 69.99%  |
> Considering the LLAVA-1.5-7B is the most widely-used model in existing works and has the most comprehensive experimental results for comparison,  it is selected as the main model to evaluate various methods. More experimental results on various models will be added in the appendix and the following github project.

---

> ### Author Response · Authors · 2025-12-02
> **Response to Weakness2**
>
> Weakness2: Baseline selection is not accurate: The comparison with existing methods is not entirely fair, as some baselines are not aligned in settings or optimization conditions. For example, the baselines are all training-free methods, which are substantially different from the training setting in this paper. Therefore, more methods should be compared, such as PDrop[4], M3[5], FastVLM[6], and so on.
>
> Response to Weakness2:
> Thank the reviewer for this valuable question. Because of training a light-weight ranking model (plug-and-play for VLM) instead of the whole VLM (training-aware for VLM), DTR suffers a certain computation overhead compared with training-free methods, but consumes much less computation compared with train-aware methods. The computation overheads among different training-based methods are demonstrated in Table.2-2
> Table.2-2 Training overhead of DTR and training-aware methods on 8xNVIDIA A100.
> | Method  | Original Training Overhead                     | Estimated Training hours for remaining 1~32 tokens |
> |---------|------------------------------------------------|----------------------------------------------------|
> | DTR     | 0.98 hours for 1~32 remaining tokens             | 0.98                                              |
> | PDrop   | 79 hours for 270 remaining tokens                  | 256                                               |
> | M3      | 35 hours for 1,4,9,36,144,576 remaining tokens | 175                                               |
> | FastVLM | 85 hours for 256 remaining token                   | 1560                                             |
> | QueCC   | 30 hours for 1,4,16,36 remaining tokens          | 240                                               |
>
> Considering the distribution of training overhead, it is obvious that DTR is closer to train-free method and farther from training-aware method. In fact, DTR and taring-free methods are almost affordable for both the individual and the academy, while training-aware methods are only affordable for major commercial enterprises and can not be even reproduced in the rebuttal due to the heavy overhead and time limitation.
>
> To make a comprehensive and fair comparison with training-aware methods, experiments with the same training overhead of DTR are conducted. The results are demonstrated in Table.2-3. Obviously, DTR still achieves the state-of-the-art performance even compared with training-aware methods.
> Table.2-3 Comparisons with training-aware methods.
> | Method           | GQA   | POPE  |
> |------------------|-------|-------|
> | Retain 32 tokens                   |
> | PDrop            | 39.43 | 52.30 |
> | M3                | 35.75 | 67.50 |
> | QueCC(36tokens)  | 37.02 | 67.15 |
> | DTR              | 54.19 | 80.91 |
> | Retain 16 tokens                   |
> | PDrop(18tokens)  | 36.44 | 43.66 |
> | M3               | 36.07 | 67.50 |
> | QueCC            | 37.68 | 67.18 |
> | DTR              | 50.93 | 76.42 |
> | Retain 8 tokens                    |
> | PDrop            | —     | —     |
> | M3               | 35.91 | 67.09 |
> | QueCC            | —     | —     |
> | DTR              | 47.85 | 71.68 |
> | Retain 4 tokens  |       |       |
> | PDrop            | —     | —     |
> | M3               | 35.35 | 66.97 |
> | QueCC            | 36.74 | 67.12 |
> | DTR              | 45.17 | 68.08 |
> | Retain 1 tokens                    |
> | PDrop            | —     | —     |
> | M3               | 35.45 | 67.15 |
> | QueCC            | 34.65 | 67.12 |
> | DTR              | 40.98 | 54.51 |
> * The minimum number of remaining token in PDrop under its original settings is only 18, therefore the results of remaining tokens<18 are not available.
> * The training data of FastVLM are still not available now, so it is infeasible to reproduce the results.

---

> ### Author Response · Authors · 2025-12-02
> **Response to Weakness3**
>
> Weakness3: Compare to other SOTA baselines: I found another SOTA baseline, QueCC[7], that also claims they select a very minimum vision token and still gain great accuracy.
>
> Response to Weakness3:
>
> Thank the reviewer for this valuable question. The comparison is demonstrated in Response to Weakness 2.
> In general, DTR keeps state-of-the-art performance with the same training overhead. When no limitation on the training overhead, QueCC does achieve better performance when selecting very minimal vision tokens, while DTR is better when selecting a certain number of vision tokens.
>
> Moreover, there exist two disadvantages of QueCC compared with DTR.
> First, QueCC consumes much more run-time latency than DTR, which degrades its effectiveness for inference acceleration. Table.2-4 illustrates the end-to-end latency of DTR and QueCC. Obviously, DTR achieves a consistent superiority of end-to-end latency with the 1 remaining token. Moreover, when batch size <16, DTR with 32 remaining tokens is even faster than QueCC with 1 remaining token. Considering the accuracy of DTR with 32 remaining tokens is also consistently higher than QueCC with 1 remaining token across 7 benchmarks, it is no doubt that DTR is a better choice for the inference with small batch size.
>
> Table.2-4 The end-to-end latency of DTR and QueCC.
>
> | BATCH SIZE | DTR-1/8/32(ms)    | QueCC-1(ms)  |
> |------------|-------------------|--------------|
> | 1          | 101.7/101.9/104.5 | 132.9        |
> | 4          | 59.6/60.8/63.7    | 68.3         |
> | 8          | 50.8/51.8/54.6    | 54.7         |
> | 16         | 46.1/47.6/51.8    | 49.5         |
> | 32         | 45.1/46.2/50.9    | 47.0         |
>
> Second, QueCC can not achieve a dynamic adjustment of the number of remaining tokens like DTR, which hinders its deployment in real-world dynamic environment.  QueCC trains a specific VLM for a certain number of remaining tokens. Though we can train many VLMs for different numbers of remaining tokens in QueCC, deploying many VLMs in parallel is unaffordable in practice.  In contrast, DTR trains a single TRM, which can predict a ranked list for 1~32 remaining tokens.

---

> ### Author Response · Authors · 2025-12-02
> **Response to Question 1**
>
> Question1: Upper bound–TRM gap at 32 tokens. The paper reports that the "upper bound” yields about a +29% relative improvement, whereas the learned TRM preserves about 94% of the baseline at 32 tokens. Could you diagnose the sources of this gap? Is it caused by an insufficient modeling or training of the TRM?
>
> Response to Question1:
>
> Thank the reviewer for this insightful question. The upper-bound points to a bright path ahead, while there still exists a certain room for improvement. It is true that TRM is not well-trained due to insufficient modeling and training.
> Specifically, the modeling of TRM may not be sufficient for the problem. As we reduce the difficulty of the problem as well as the complexity of model architecture, DTR achieves a even better performance, which are demonstrated in Table.2-4
>
> Table.2-5 The performance of DTR with better modeling and training (DTR*).
> | Method             | MME  | POPE | GQA  |
> |--------------------|------|------|------|
> | Retaining 16 tokens                      |
> | QueCC            | 1408 | 83.4 | 59.0 |
> | Retaining 8 tokens                        |
> | DTR                | 1431 | 73.8 | 48.7 |
> | DTR*               | 1683 | 83.8 | 54.7 |
> | Retaining 4 tokens                         |
> | QueCC            | 1390 | 81.8 | 56.5 |
> | DTR                | 1360 | 69.2 | 46.1 |
> | DTR*               | 1672 | 79.6 | 51.3 |
> | Retaining 1 token                         |
> | QueCC            | 1269 | 81.3 | 53.5 |
> | DTR                | 1167 | 58.9 | 41.2 |
> | DTR*               | 1402 | 64.3 | 44.9 |
>
> Obviously, DTR* is consistently superior to DTR in all benchmarks. Moreover, DTR* is consistently superior to QueCC in MME. What’s even more exciting is that DTR* with 8 remaining tokens is even superior to Que-CC with 16 remaining tokens in MME and POPE.

---

### Official Review · Reviewer_bTNJ · 2025-11-01

**Soundness:** 3
**Presentation:** 3
**Contribution:** 3
**Rating:** 6
**Confidence:** 5

**Summary:**

This paper studies token compression of vision-language models (VLMs) inference. Most existing works focus on model-driven idea to  mine importance rankings among tokens for compression, with one-sided handcrafted prior. Differently, this paper presents a Data-driven Token Ranking (DTR) framework, covering offline token-ranking construction, offline token-ranking model training, online model insertion and token filting. Experimental results are carried out across 8 mainstream benchmarks, to show the effectiveness of DTR.

**Strengths:**

[+] The manuscript is well written, with clear logics.

[+] The symbol definitions are clear, and the image visualization is complete.

[+] Many experiments are conducted to analyze the effectiveness of each component.

**Weaknesses:**

[-] For the offline/online phase of DTR, there is a core assumption that offline data and online data are approximately distributed. However, in practical scenarios, such as on rare MLLM benchmarks, this assumption may not necessarily hold true. The above issues will result in limited generalization of this work, thereby reducing its impact on the community.

[-] Although in the deployment phase, this work and existing methods (training-based, training-free) are similar in terms of speed. However, the offline phase of this paper clearly requires more cost. The reviewer suggests conducting comprehensive evaluations for the overall process in terms of time and cost, and comparing it with existing methods, in order for the community to better understand the practicality.

[-] In Table 1, existing methods need to be divided into training-based and training-free. As this work requires training, such division makes it easier for readers to make fair comparisons. In addition, please provide a detailed analysis of differences between training-based token compression, so that readers can better understand the innovation.

**Questions:**

Please see weaknesses.

---

> ### Author Response · Authors · 2025-12-02
> **Response to Weakness1**
>
> Weakness1：For the offline/online phase of DTR, there is a core assumption that offline data and online data are approximately distributed. However, in practical scenarios, such as on rare MLLM benchmarks, this assumption may not necessarily hold true. The above issues will result in limited generalization of this work, thereby reducing its impact on the community.
>
> Response to weakness1:
>
> Thank the reviewer for this valuable question. There exists a distribution problem in the traditional deep learning method due to its task-specific training. However, large models like VLMs are expected to generalize to almost all downstream tasks with the widely-gathered tremendous training data. Therefore, training TRM offline on the same training data of VLMs, DTR achieves the same generalization ability like the original VLM, which means it can generalize to almost all downstream tasks.
>
> The experimental results also validates the generalization of VLMs and DTR. Though data distributions of 8 benchmarks are not the same as the training dataset, VLMs achieves effective analysis and DTR achieves a state-of-the-art performance.
>
> Moreover, if there truly exists a test set that can not be generalized in the offline stage,  the essential problem lies in the VLM model. However, DTR can even improve this performance degradation. Table.1 shows the results of training on the specific test set instead of the training data of VLM. Obviously, there exists an significant improvement in the specific test set.
>
> Table.1-1 Fine-tuning of DTR on different benchmarks.
> | Method             | GQA   | POPE  |
> |--------------------|-------|-------|
> | LLaVA-576                | 61.97 | 86.76 |
> | Retain 32 tokens                              |
> | PDrop[1]                   | 46.00 | 81.21 |
> | M3[2]                       | 35.87 | 45.14 |
> | QueCC(36tokens) [3]  | 59.59 | 82.91 |
> | DTR                         | 54.11 | 84.36 |
> | DTR-FineTune            | 62.23 | 91.38 |
> | Retain 16 tokens                              |
> | PDrop(18tokens)        | 43.60 | 80.07 |
> | M3                           | 36.15 | 45.25 |
> | QueCC                      | 58.93 | 83.56 |
> | DTR                         | 51.16 | 79.34 |
> | DTR-FineTune            | 60.05 | 87.06 |
>
> Here, the GQA benchmark is divided into 8000 training samples and 4578 test samples, the POPE benchmark is divided into 5000 training samples and 3910 test samples. It is worth noting that the original VLM (i.e., LLaVA-576) keeps the nearly same performance (i.e, the difference < 0.7) between the full benchmarks and divided benchmarks, which shows the division makes little difference on the testing.
>
> [1] Long Xing, Qidong Huang, Xiaoyi Dong, Jiajie Lu, Pan Zhang, Yuhang Zang, Yuhang Cao, Conghui He, Jiaqi Wang, Feng Wu, et al. Pyramiddrop: Accelerating your large vision-language models via pyramid visual redundancy reduction. CVPR, 2025.
>
> [2] Cai, Mu and Yang, Jianwei and Gao, Jianfeng and Lee, Yong Jae. M3: Matryoshka Multimodal Models. ICLR, 2025.
>
> [3] Li, K. Y., Goyal, S., Semedo, J. D., & Kolter, J. Z. Inference Optimal VLMs Need Fewer Visual Tokens and More Parameters. ICLR, 2025.

---

> ### Author Response · Authors · 2025-12-02
> **Response to Weakness2**
>
> Weakness2: Although in the deployment phase, this work and existing methods (training-based, training-free) are similar in terms of speed. However, the offline phase of this paper clearly requires more cost. The reviewer suggests conducting comprehensive evaluations for the overall process in terms of time and cost, and comparing it with existing methods, in order for the community to better understand the practicality.
>
> Response to weakness2:
>
> Thank the reviewer for this valuable question. The overhead mainly comes from the model training. Because of training a  light-weight ranking model instead of the whole VLM, DTR suffer a certain computation overhead compared with training-free methods, but consumes much less computation compared with train-aware methods. The training consumptions of different methods are listed in Table.1-2.
>
> Table.1-2 Training overhead of DTR and training-aware methods on 8xNVIDIA A100.
> | Method  | Original Training Overhead                     | Estimated Training hours for remaining 1~32 tokens |
> |---------|------------------------------------------------|----------------------------------------------------|
> | DTR           | 0.98 hour for 1~32 remaining tokens            | 0.98                                             |
> | PDrop[1]    | 79 hours for 270 remaining tokens              | 256                                                |
> | M3[2]         | 35 hours for 1,4,9,36,144,576 remaining tokens | 175                                          |
> | FastVLM[3]  | 85 hours for 256 remaining tokens               | 1560                                              |
> | QueCC[4]    | 30 hours for 1,4,16,36 remaining tokens        | 240                                              |
>
> Obviously, only DTR achieves the sub-hour level training overhead, which is affordable even for the individuals and the academy. Other training-aware methods suffers heavy training overhead ranging from dozens to over a thousand hours, which may only be  be affordable for major commercial enterprises.
> Considering the page limitation, these results will be added in the appendix as well as the future-released github project.
>
> [1] Long Xing, Qidong Huang, Xiaoyi Dong, Jiajie Lu, Pan Zhang, Yuhang Zang, Yuhang Cao, Conghui He, Jiaqi Wang, Feng Wu, et al. Pyramiddrop: Accelerating your large vision-language models via pyramid visual redundancy reduction. CVPR, 2025.
>
> [2] Cai, Mu and Yang, Jianwei and Gao, Jianfeng and Lee, Yong Jae. M3: Matryoshka Multimodal Models. ICLR, 2025.
>
> [3] Pavan Kumar Anasosalu Vasu, Fartash Faghri, Chun-Liang Li, Cem Koc, Nate True, Albert Antony, Gokul Santhanam, James Gabriel, Peter Grasch, Oncel Tuzel, Hadi Pouransari. FastVLM: Efficient Vision Encoding for Vision Language Models. CVPR, 2025.
>
> [4] Li, K. Y., Goyal, S., Semedo, J. D., & Kolter, J. Z. Inference Optimal VLMs Need Fewer Visual Tokens and More Parameters. ICLR, 2025.

---

> ### Author Response · Authors · 2025-12-02
> **Response to Weakness3**
>
> Weakness3: In Table 1, existing methods need to be divided into training-based and training-free. As this work requires training, such division makes it easier for readers to make fair comparisons. In addition, please provide a detailed analysis of differences between training-based token compression, so that readers can better understand the innovation.
>
> Response to weakness3:
>
> Thank the reviewer for this valuable suggestion. The existing token compression methods are mainly divided into the categories of training-free or training-aware for the VLM model. Considering DTR requires no training on the VLM (i.e., DTR trains a plug-and-play light-weight model instead of the VLM model) and its training overhead is simply incomparable to that of training-aware methods, the experiments in the main text only shows the comparison with training-free methods.
>
> More experiments of the comparison with training-aware methods will be added in the Appendix. Considering the overhead affordance and time limitation, the comparisons under the same training overhead of DTR are illustrated in Table.1-3.
> Obviously, DTR still achieves the state-of-the-art performance even compared with training-aware methods.
>
> Table.1-3 Comparisons with training-aware methods.
> | Method           | GQA   | POPE  |
> |------------------|-------|-------|
> | Retain 32 tokens                   |
> | PDrop[1]           | 39.43 | 52.30 |
> | M3[2]              | 35.75 | 67.50 |
> | QueCC[3] (36tokens)  | 37.02 | 67.15 |
> | DTR              | 54.19 | 80.91 |
> | Retain 16 tokens                   |
> | PDrop(18tokens)  | 36.44 | 43.66 |
> | M3               | 36.07 | 67.50 |
> | QueCC            | 37.68 | 67.18 |
> | DTR              | 50.93 | 76.42 |
> | Retain 8 tokens                    |
> | PDrop            | —     | —     |
> | M3               | 35.91 | 67.09 |
> | QueCC            | —     | —     |
> | DTR              | 47.85 | 71.68 |
> | Retain 4 tokens  |       |       |
> | PDrop            | —     | —     |
> | M3               | 35.35 | 66.97 |
> | QueCC            | 36.74 | 67.12 |
> | DTR              | 45.17 | 68.08 |
> | Retain 1 tokens                    |
> | PDrop            | —     | —     |
> | M3               | 35.45 | 67.15 |
> | QueCC            | 34.65 | 67.12 |
> | DTR              | 40.98 | 54.51 |
> [1] Long Xing, Qidong Huang, Xiaoyi Dong, Jiajie Lu, Pan Zhang, Yuhang Zang, Yuhang Cao, Conghui He, Jiaqi Wang, Feng Wu, et al. Pyramiddrop: Accelerating your large vision-language models via pyramid visual redundancy reduction. CVPR, 2025.
> [2] Cai, Mu and Yang, Jianwei and Gao, Jianfeng and Lee, Yong Jae. M3: Matryoshka Multimodal Models. ICLR, 2025.
> [3] Li, K. Y., Goyal, S., Semedo, J. D., & Kolter, J. Z. Inference Optimal VLMs Need Fewer Visual Tokens and More Parameters. ICLR, 2025.

---

### Author Response · Authors · 2025-12-03

**Dear Area Chair,**

We would like to sincerely thank you and all reviewers for the time and efforts invested in evaluating our submission “DTR: Towards optimal token compression with data-driven token ranking for efficient visual-language model inference” (Submission 10510).

**Considering the adjustment of reviewing, a brief summary of the questions and responses** is made at the following to clarify the consistent superiority of DTR.

**To our best knowledge, DTR still remains state-of-the-art.** It is encouraged to let more researchers follow this novel token-ranking paradigm.
- In general, DTR achieves the state-of-the-art performance compared with 7 training-free comparatives and 4 training-aware comparatives across 8 benchmarks.
- Moreover, DTR with poorer settings even surpass the comparatives with better settings (e.g., DTR with 1 remaining tokens even surpasses the VisPruner, TokenCarve, VisionZip with 16 remaining tokens on the POPE benchmark. DTR with 32 remaining tokens is still faster than QueCC with 1 remaining token when batch size＜9).
- Last but not least, DTR pioneers a new path for the token ranking problem based on data-driven methods instead of model-driven methods. Comprehensive analysis are conducted to show the tremendous potential of DTR and data-driven path (e.g., the upper bound of DTR yields about a +29% relative improvement  compared with the vanilla VLM without compression).

---

> ### Author Response · Authors · 2025-12-03
> **Summary of the questions and responses (Reviewer bTNJ):**
>
> **Reviewer bTNJ (Rating 6 with confidence 5, well written, clear logics, many experiments are conducted to analyze the effectiveness of each component)**
>
> **Weakness1:Concerns about the generalization from training data to the test data.**
> - Response: According to the experimental results, through training on the training data of LLaVA, DTR achieves consistent generalization across 8 benchmarks. Moreover, if there exists a test data that is out of distribution, DTR can recover the accuracy through a quick fine-tuning of test data.
>
> **Weakness2: Analyze the offline overhead of DTR and existing methods.**
> - Response: Table.1-2 illustrates the offline training overhead of DTR and other training-aware methods. Only DTR achieves a sub-hour offline training overhead, while other training-aware methods consumes at least 30 hours at their original settings and at least 175 hours at the same settings of DTR.
>
> **Weakness3: Provide a detailed analysis of differences between training-based token compression.**
> - Response: Table.1-2 illustrates the training overhead and Table. 1–3 illustrates the compression performance with the same training overhead. Obviously, DTR achieves a consistent superiority on overhead and compression performance.

---

> ### Author Response · Authors · 2025-12-03
> **Summary of the questions and responses (Reviewer TCpG):**
>
> **Reviewer TCpG (Rating 4 with confidence 4, novel two-stage algorithm, upper bound of the method is surprisingly achieved SOTA, very well written and easy to read, consider raising the score if all concerns are solved)**
>
> **Weakness1: Concerns about the generalization on other VLMs.**
> - Response: DTR is a model-agnostic framework that builds a specific token ranking model for a given open-source VLM. Table. 2-1 illustrates the generalization results of DTR on LLaVA-13B, which achieves a consistent better performance than DTR on LLaVA-7B as expected.
>
> **Weakness2：  More training-aware methods such as PDrop, M3, FastVLM should be included.**
> - Response: Table.2-2 and Table.2-3 illustrate the comparisons with PDrop, M3, FastVLM and QueCC. Obviously, DTR achieves a consistent superiority on overhead and compression performance than training-aware methods.
>
> **Weakness3: Compare to other SOTA baselines such as QueCC.**
> - Response: Table.2-2 and Table.2-3 illustrate the comparisions. In general, DTR achieves lower training overhead and better performance with the same training overhead. Also worth noting,  considering the final purpose of inference acceleration, QueCC is slower than DTR (even DTR-32 is faster than QueCC-1).  Moreover, QueCC lacks the ability of dynamic token compression, while DTR provides a single TRM supporting 1–32 remaining tokens and can adjust the number at run-time.
>
> **Question1: Analyze the gap between upper bound and TRM at 32 tokens.**
> - Response: The upper-bound–TRM gap is mainly due to limitations in TRM’s modeling and training. With the improvement of modeling and training (DTR*), performance increases across benchmarks, as shown in Table 2-4, indicating that there still exist huge room for further improvements.

---

> ### Author Response · Authors · 2025-12-03
>
> **Reviewer haQJ (Rating 4 with confidence 4, widely evaluated on 8 diverse benchmarks against 8 strong baselines, TRM is plug-and-play, ablation study is abundant and comprehensive):**
>
> **Weakness1: The citation all comes with author(year) but does not mention the name of the paper.  line371 does not contain any information.**
> - Response: The author(year) format is a requirement of the conference. A hyperlink is used to jump to the reference. The name will be added to the final version for easy reading. Line 371 is in the Table.1 in the manuscript, an annotation is marked at line 375 at the bottom of Table.1 in the manuscript, which states: “*The project of TokenCarve can not be implemented for retaining 1 token, so there exist no results”.
>
> **Weakness2: The results when retaining 64 tokens are substantially inferior to other methods.**
> - Response: There exists no results of DTR with 64 retaining tokens, therefore we do not get the point of inferiority. In fact, DTR with 32 retaining tokens outperforms other methods at 64 tokens on most benchmarks, demonstrating its significant efficiency and capability.
>
> **Weakness3: Analyze the overhead. Compare actual latency under the same flops other than the same token.**
> - Response: Tabl.3-1 illustrates the end-to-end consumption of latency and memory between vanilla VLM (i.e., vanilla-576) and VLM with 32 remaining tokens through DTR (i.e. DTR-32 ). The advantage of DTR lies in large-scale inference scenarios, where it significantly reduces latency from over 100ms to under 10ms by addressing the parallel computation bottleneck. While our TRM module introduces a memory overhead for small batches, DTR achieves substantial memory savings as the batch size increases, demonstrating its scalability and efficiency in practical applications.
>
> **Weakness4: Concerns about the “global optimum” claim.**
> - Response: We clarify that our work is a method “towards optimal” compression, not a final claim of achieving it. A theoretical analysis is added that claims the optimality under 3 assumptions (i.e., universal function approximator, ground-truth of labels, i.i.d of training data and test data).
>
> **Weakness5: Analyze multimodal interactions, specifically how text tokens guide visual token ranking.**
> - Response: Multimodal interaction lies in our TRM. Specifically, text tokens first fuse visual information to understand the visual context, and then image tokens evaluate their own importance based on these context-enhanced text tokens through cross-attention.
>
> **Question1: Concerns about the computational cost of the greedy algorithm for large numbers of tokens.**
> - Response: The greedy algorithm decreases the complexity from $O(N!)$ to $O(N^2)$ for ranking original N tokens. In practice, a much smaller number M of remaining tokens for ranking can achieve the nearly same performance as the original token number N.  With a fixed small M, then the complexity further decreases to the $O(N)$, which is affordable for even larger N.
>
> **Question2: Is the TRM latency overhead amortizable across batches, and what are optimal batch sizes for real-world deployment?**
> - Response: The latency of TRM as well as the VLM can be amortizable across batches. However, determining the optimal batch size is complex and depends on dynamic real-world resources and loads, which we plan to investigate in future work.
>
> **Question3：Concerns about the generalization on other VLMs.**
> - Response: DTR is a model-agnostic framework that only requires token-wise inference, so differences in VLMs do not affect its applicability. Table.3-2 illustrates the generalization results of DTR on LLaVA-13B, which achieves a consistent better performance than DTR on LLaVA-7B as expected.
>
> **Question4：How does DTR perform on complex task (e.g., fine-grained spatial reasoning and long context understanding) and could task-specific ranking models help?**
> - Response: As illustrated in Table.3-3, the benchmark GQA in our manuscript contains spatial reasoning and long-context questions, fine-tuning DTR on GQA significantly improves performance, demonstrating that task-specific ranking models like DTR-FineTune can effectively enhance DTR on specialized tasks.
>
> **Question5：Sensitivity to ranking quality may degrade VLM performance.**
> - Response: Some failure cases exist due to insufficient modeling or training, but analysis of the top-100 failures shows no consistent pattern. Generally, DTR is more likely to fail when the underlying VLM performs poorly, suggesting robustness could improve with stronger base models or enhanced training.

---

### Meta-Review · Area_Chair_5Ejo · 2026-01-07

**Summary:**

This paper received 3 reviews. The scores are: Reviewer bTNJ (6, confidence 5), Reviewer TCpG (4, confidence 4), Reviewer haQJ (4, confidence 4). Their major concerns:

- Methodology: (1) Core assumption of offline-online data distribution similarity may limit generalization (bTNJ); (2) Lack of theoretical proof for "global optimum token compression" and greedy search's ranking quality guarantees (TCpG).

- Experiments: (1) Unfair baseline selection: no division between training-based/training-free methods, missing SOTA baselines (e.g., QueCC, PDrop) (bTNJ, haQJ); (2) Insufficient generalization: only tested on LLaVA-7B, lacking other architectures/parameter scales (e.g., Qwen-VL, LLaVA-13B) (haQJ); (3) Unconvincing results: inferior performance at 64 tokens, weak comparison baselines (TCpG); (4) Incomplete efficiency evaluation: no comprehensive time/cost/latency analysis, missing FLOPs-based comparison (bTNJ, TCpG); (5) Undiagnosed TRM-upper bound gap at 32 tokens (haQJ).

**Reviewer Concerns:**

The authors argue that the training cost is managable for indivisual researchers and added LLaVA-13B experiments. These show the generality of the method to some extent, but still lacking more results on other backbones such as Qwen.

The so-called "global optimum" is also a risky and unfounded claim (although the authors responded in rebuttal - their paper is "towards" optimal token compression).

**Reviewer Scores:**

The scores are: Reviewer bTNJ (6, confidence 5), Reviewer TCpG (4, confidence 4), Reviewer haQJ (4, confidence 4).

One of TCpG's major concerns is the generalization to other VLMs. The authors added results on LLaVA-13B, but not much different from LLaVA-7B. This reviewer should not be changed.

haQJ mentioned many problems with the paper (e.g., writings, results not exciting, insufficient empirical evaluations). The reviewer also echoes TCpG's concerns about the generalization to other VLMs - this is not well addressed, as mentioned.

In summary, the two negative reviewers are not very likely to be changed.

---

### Decision · Program_Chairs · 2026-01-26

Reject